# Mechanical forces drive ordered patterning of hair cells in the mammalian inner ear

Roie Cohen[1,2,3,9], Liat Amir-Zilberstein[1,9], Micha Hersch[4,5], Shiran Woland[1], Olga Loza[1], Shahar Taiber [1,6], Fumio Matsuzaki [7], Sven Bergmann [4,5,8], Karen B. Avraham [6] & David Sprinzak[1,2✉]

Periodic organization of cells is required for the function of many organs and tissues. The development of such periodic patterns is typically associated with mechanisms based on intercellular signaling such as lateral inhibition and Turing patterning. Here we show that the transition from disordered to ordered checkerboard-like pattern of hair cells and supporting cells in the mammalian hearing organ, the organ of Corti, is likely based on mechanical forces rather than signaling events. Using time-lapse imaging of mouse cochlear explants, we show that hair cells rearrange gradually into a checkerboard-like pattern through a tissue-wide shear motion that coordinates intercalation and delamination events. Using mechanical models of the tissue, we show that global shear and local repulsion forces on hair cells are sufficient to drive the transition from disordered to ordered cellular pattern. Our findings suggest that mechanical forces drive ordered hair cell patterning in a process strikingly analogous to the process of shear-induced crystallization in polymer and granular physics.

[1] George S. Wise Faculty of Life Sciences, School of Neurobiology, Biochemistry and Biophysics, Tel Aviv University, 6997801 Tel Aviv, Israel. [2] The Center for Physics and Chemistry of Living Systems, Tel Aviv University, 6997801 Tel Aviv, Israel. [3] Faculty of Exact Sciences, Raymond and Beverly Sackler School of Physics and Astronomy, Tel Aviv University, 6997801 Tel Aviv, Israel. [4] Department of Computational Biology, University of Lausanne, 1015 Lausanne, Switzerland. [5] Swiss Institute of Bioinformatics, 1015 Lausanne, Switzerland. [6] Sackler Faculty of Medicine and Sagol School of Neuroscience, Department of Human Molecular Genetics and Biochemistry, Tel Aviv University, 6997801 Tel Aviv, Israel. [7] Laboratory of Cell Asymmetry, RIKEN Center for Biosystems Dynamics Research, Kobe 650-0047, Japan. [8] Department of Integrative Biomedical Sciences, University of Cape Town, Cape Town, South Africa. [9]These authors contributed equally: Roie Cohen, Liat Amir-Zilberstein. ✉email: davidsp@tauex.tau.ac.il

The mature organ of Corti is a strip of epithelial cells that extends along the base-to-apex axis within the cochlear duct (Fig. 1a, b) which comprises of exactly four rows of hair cells (HCs), three rows of outer hair cells (OHCs) and one row inner hair cells (IHCs), regularly interspersed by non-sensory supporting cells (SCs). The OHC region is flanked by the Hensen cells on its lateral side and a single row of cuboidal-shaped inner pillar cells that separate it from the IHCs on its medial side. Studies in mice showed that the differentiation into HCs and SCs initiates from a disordered undifferentiated layer of cells at around embryonic day 14 (E14), and involves a lateral inhibition process mediated by Notch signaling[1–3]. At the same time, the organ of Corti exhibits significant morphological changes that include convergent extension[4–6] and cell movements[7]. No significant cell division or cell death events are observed within the organ of Corti during this process[4,8].

## Results

**The organ of Corti gradually organizes into an ordered pattern**. To determine the processes that drive the transition from a disordered undifferentiated tissue to an ordered pattern of HCs and SCs we first quantitatively analyzed the morphological changes that occur in the organ of Corti in space and time. We utilized the fact that the organization of the organ of Corti exhibits a developmental gradient along the base-to-apex axis[1], allowing continuous analysis of the patterning progression in fixed samples. To this end, we imaged fixed cochleae at different developmental stages (E15.5 to postnatal day 0 (P0)) taken from transgenic mice expressing Math1-GFP[9], which is an early marker of HC differentiation. The cochleae were additionally immunostained with an antibody against zonula occludens-1 (ZO1), a known marker of tight junctions[10], delineating cell boundaries at the apical side of the tissue (Fig. 1b–d). Images of full cochleae were reconstructed by tiling multiple high-resolution confocal images taken at different Z-planes. Images were then segmented and analyzed using a custom-built code (Fig. S1a) allowing extraction of morphological parameters (e.g. cell size, number of neighbors) and cell identity (HC or SC).

The analysis both along the base-to-apex axis and at different time points demonstrates the gradual differentiation and organization of the organ of Corti in both space and time (Fig. 1c). To obtain a quantitative measure of the level of organization we defined two local order parameters. The first parameter is the number of SC neighbors for each HC in the OHC region. In the final pattern, each OHC from the middle HC row (OHC$_2$) has almost always four SC neighbors, while at earlier stages this number is often higher (Fig. 1d, middle row). Analysis of cochleae from E15.5, E17.5, and P0 shows that the average number of SC neighbors decreases as development progresses (both in space and time, Fig. 1e). In a complementary manner, the number of HC neighbors for SCs from the middle row increases with the developmental stage (Fig. S1b).

The second-order parameter defines a measure for the hexagonal packing of HCs. In a fully ordered organ of Corti the HCs are organized in a deformed monoclinic crystal structure along the spiral-shaped cochlea (Fig. S1c, d). Each HC from OHC$_2$ is at the center of a regular hexagon stretched in one axis, formed by its closest HC neighbors (Fig. 1d, bottom row). To obtain a measure of the local hexagonal order, we use a modified version of the common hexagonal order parameter $\psi_6$[11]. This parameter value is one for a perfect regular hexagon and close to zero for an uncorrelated set of points. To measure the ordered organization of HCs, we used a modified order parameter, termed $\psi_6^*$, which takes into account the stretching of the pattern along the base-to-apex axis. $\psi_6^*$ was calculated for the centroids of

neighboring HCs of each cell from OHC$_2$ by first estimating the degree of stretching and then calculating $\psi_6$ for the scaled centroid positions (see methods, from here on we refer to higher values of $\psi_6^*$ as higher hexagonal order). Analysis across all the cochleae measured, showed that the hexagonal order parameter, $\psi_6^*$, increases in value with the developmental stage, indicating the gradual organization of the HCs into a slightly stretched hexagonal pattern (i.e. monoclinic, Fig. 1f). To further demonstrate the emergence of crystalline order of HCs, we calculate the structure factor using the HCs centroids after compensating for cochlear curvature (see methods). Clear Bragg peaks emerge with developmental time, further indicating the gradual organization of the HCs (Fig. S1e). Consistent with earlier works[12], we also find that the apical surface areas of the HCs increase with the developmental stage, while the corresponding areas of the SCs slightly decrease (Fig. 1g and Fig S2a, b). Overall, this analysis shows that gradual transition into an organized pattern involves a decrease in the number of SC neighbors of each HC, an increase in the hexagonal order of HCs, and an increase in the relative areas of HCs with respect to that of SCs.

**Shear motion is observed during organization of HCs and SCs**. To understand the dynamics underlying the morphological changes that take place during the development of the organ of Corti, we developed an assay for live imaging of inner ear explants. In this assay, cochlear explants from transgenic mice expressing the boundary marker ZO1-EGFP[13] were imaged at high resolution using a confocal microscope equipped with an Airyscan detector, for up to 24 h. Movies of cochlear explants were performed on cochleae extracted at E15.5, E17.5, and P0.

Remarkable reorganization processes were observed at earlier developmental stages. Imaging of E15.5 explants, showed a significant shear motion of the HCs in the direction of cochlear extension (Fig. 2a, b, Supplementary Video 1). Tracking single cells in these movies allowed us to measure the relative displacement of each cell as a function of time. Furthermore, the cells can be classified into apparent HCs and SCs based on their morphology at the final frame of the movie, where apparent HCs are more convex and apparent SCs are more concave. Analysis of the relative displacement of the lateral and medial parts of the OHC region, showed that cells farther away from the pillar cell layer (dashed line in Fig. 2a) exhibited larger displacement than cells closer to the pillar cell layer (Fig. 2c top, Fig. S3a left). In the medial-lateral axis (perpendicular to the base-to-apex axis) HCs and SCs showed different tendencies. While HCs tended to stay at the same lateral position or slightly converge, the SCs tended to move laterally (Fig. 2c bottom, Fig. S3a right). We also note that the Hensen cells, lateral to the OHCs, are highly dynamic in their shape and exhibit directional sliding movement with respect to the OHC layer even at more advanced developmental stages (Fig. S3b and Supplementary Video 2). Overall, these observations suggest that OHCs and SCs in the OHC region undergo shear motion, potentially driven by directional sliding movement of Hensen cells, while in the lateral direction, SCs move laterally with respect to the OHCs.

**HCs and SCs undergo intercalations and delaminations**. Previous studies of developing epithelial tissues showed that global morphological changes often involve local morphological transitions including cellular intercalations (T1 transitions) and delaminations (T2 transitions)[14,15]. Analysis of our movies indeed showed both intercalations and delaminations both at E15.5 (Fig. 2d, e and Supplementary Videos 3 and 4) and at E17.5 (Fig. S3c, d and Supplementary Videos 5 and 6). We find that the number of intercalations observed is significantly higher at earlier

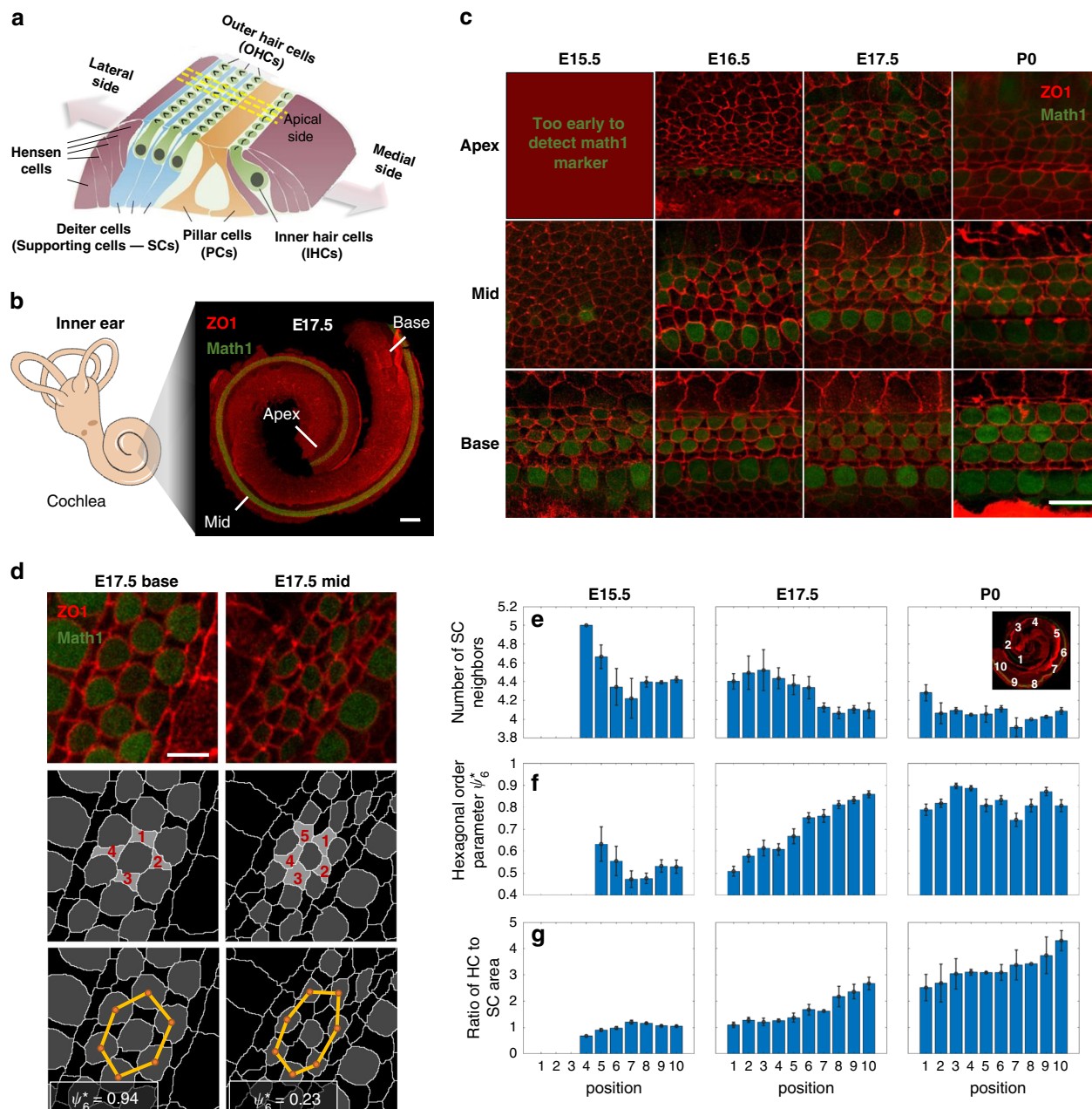

**Fig. 1 Hair cells gradually reorganize into a checkerboard-like pattern. a** Schematic of the organ of Corti. **b** Left: schematic of the mammalian inner ear. Right: Confocal image of the cochlea of a E17.5 mouse embryo marking the base, mid and apex regions. Cochleae are taken from transgenic mice expressing Math1-GFP (green) and are immunostained with α-ZO1 (red). Scale bar: 100 μm. **c** Representative images at different developmental times and different positions along the base-to-apex axis. Rows and columns correspond to different positions along the cochlear axis and different development times, respectively, as indicated. Scale bar: 10 μm. **d** Schematic of the definition of two order parameters: (i) The number of SC neighbors of each HC in $OHC_2$ (middle row) and (ii) hexagonal order parameter $\psi_6^*$ (bottom row). Yellow lines connecting HC centroids (orange dots) demonstrate higher hexagonal order at the base relative to the mid. $\psi_6^*$ values for each centroid cluster are as indicated. Scale bar: 5 μm. **e–g** Morphological and order parameters in different regions of the cochlea from apex to base (defined in inset) and at different developmental times (columns). Rows correspond to number of SCs neighbors (**e**), hexagonal order parameter $\psi_6^*$ (**f**), and ratio of HC to SC surface area (**g**). Local measures of order parameters associated with each HC are pooled by developmental age over $n = 3,4,3$ cochleae at E15.5, E17.5, P0, respectively, and then binned by cochlear position. Bars represent average on all local orders parameters within each bin. Error bars represent S.E.M. Schematic in **a** is modified with permission from Dror and Avraham[34].

developmental stages (E15.5) compared to later developmental stages (E17.5) (Fig. 2f), and inversely correlated with the degree of tissue organization. The number of delaminations is relatively small, typically between 0-2 delaminations per field of view per 8 h movie.

A specific type of reorganization observed in our movies, is a situation where a cell, which is initially located within the middle of the OHC region, is "squeezed" laterally (Fig. 2g and Supplementary Video 7). Such lateral squeezing processes likely drive the lateral movement of SCs shown in Fig. 2c. Both lateral squeezing and cellular delaminations seem to be restricted to cells

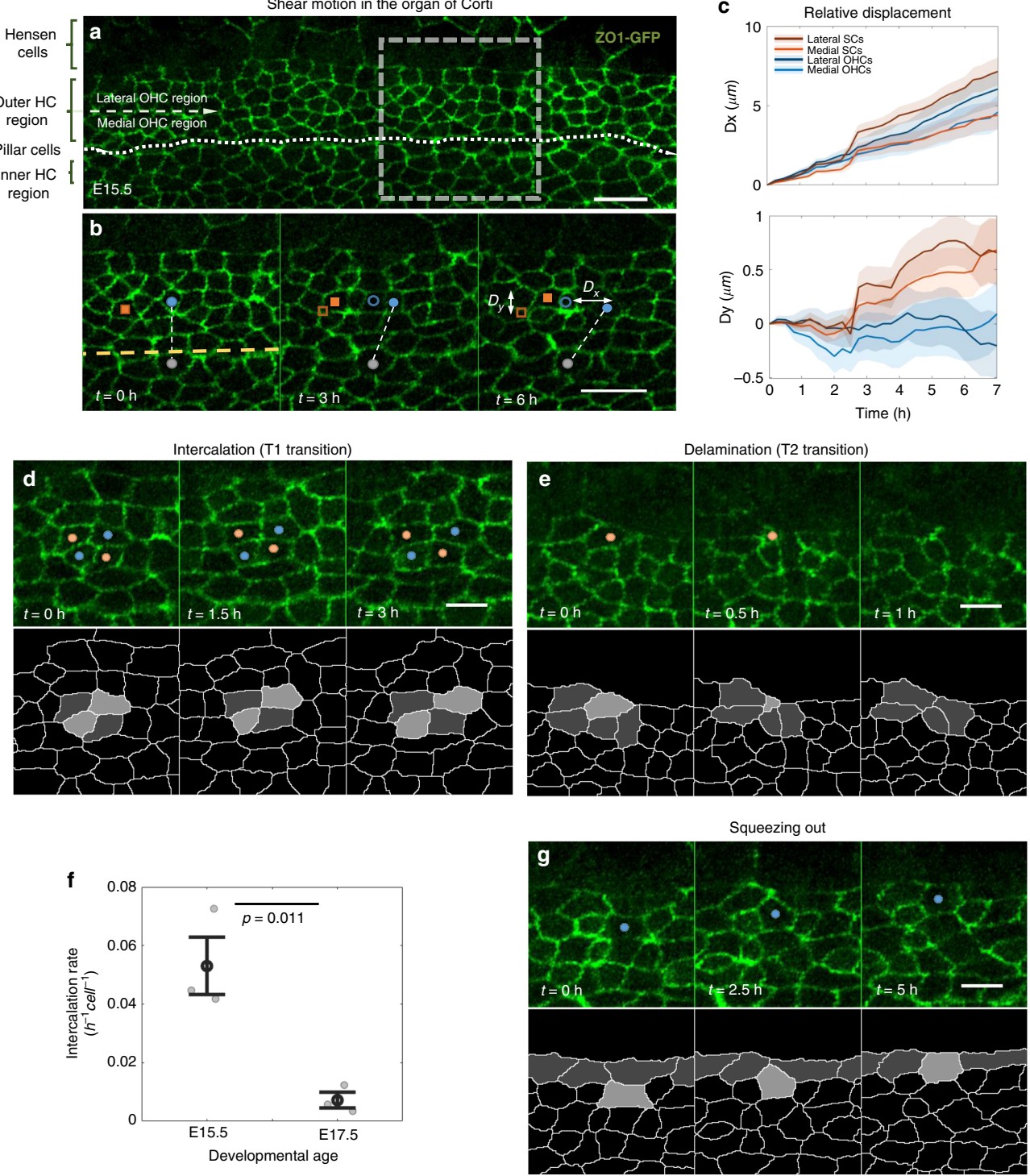

**f** Intercalation rate versus developmental age.

with SC morphology (more concave shaped). Both of these processes may contribute to the reduction in the number of SC neighbors of HCs shown in Fig. 1e.

**Mechanical model captures main aspects of inner ear development.** To get an insight into the processes that lead to ordered patterning of the organ of Corti, we developed a mechanical model of the patterning process. As a first stage, we wanted to determine the minimal rules that can capture the transition from a disordered epithelial layer to an ordered checkerboard-like pattern, focusing on the OHC patterning (Fig. 3a). We chose to use a

modified 2D vertex model, previously used to describe morphological transitions in growing epithelial tissues[15,16]. In 2D vertex models, different cells are attributed with different mechanical properties associated with the cell area, cell perimeter, and the cellular junctions, while the cellular configuration is determined by minimizing the overall mechanical energy of the system (Fig. S4a). For the model of the organ of Corti, each cell is defined as a HC, SC, boundary cells (e.g. pillar cells and top boundary cells), or a general cell outside the organ of Corti (Fig. 3a). In parallel to the minimization of the mechanical energy we introduced in the model intercalations and delaminations that facilitate morphological changes. Intercalations and delaminations initiate when the

**Fig. 2 Shear motion and morphological transitions drive organization in the organ of Corti. a, b** A mid-apex region of a cochlear explant of a ZO1-EGFP mouse at E15.5. **a** Image showing the different regions along the medial-lateral axis of the Organ of Corti. Dotted line marks the pillar cell row. Dashed arrow separates the medial and lateral regions of the OHC region. **b** A filmstrip of the dashed square region in **a**, showing shear motion in the OHC region. A connecting line between an OHC (blue dot) and an IHC (gray dot) highlights the relative motion between the cells. The displacement of a HC (blue dot) and a SC (red square) is indicated by the distance from their initial positions (blue empty circle and red empty square, respectively). $D_x$, $D_y$ are the total displacements in the x and y directions at the end of the movie compared to the initial position. Scale bars: 10 μm. Movie shown in Supplementary Video 1. **c** Displacement of apparent HCs and SCs from the movie shown in **b**. Displacements are calculated relative to the initial position of each cell. Cells from the medial (light red, light blue) and lateral (dark red, dark blue) OHC regions display different motion profiles. Shaded regions represent the boundaries of S.E.M. **d, e** Filmstrips showing **d** an intercalation process between two cell pairs (marked with red and blue dots), and **e** a delamination process of the cell marked with red dot. Bottom rows present segmented versions of the transitions. Movies shown in Supplementary Videos 3 and 4, respectively. **f** Rate of intercalations in the organ of Corti at E15.5 and E17.5. Gray dots correspond to individual data points obtained from $n = 3$ movies. Black dots and error bars represent average and S.E.M, respectively. Statistical analysis was done using two-sided two-sample t-test. **g** A filmstrip showing an event where a cell (blue dot) is "squeezed out" toward the top border of the OHC region. Bottom row presents a segmented version of the process. Movie shown in Supplementary Video 7. Scale bars for **d**, **e**, **g**: 5 μm. Observations in **a**, **b**, **d**, **e**, **g** were seen in all three repeats.

junction or cell area falls below some threshold length or threshold area, respectively. For simplicity, we use periodic boundary conditions at the edges of the cell lattice.

Since we focus on the differentiation and patterning of the three OHC rows, and since the IHCs and pillar cells appear prior to the OHC differentiation (Fig. 1c), we start the simulations with the IHCs and the adjacent rectangular-shaped pillar cell row already formed (Fig. 3a, left). To simulate the initial alternating pattern of OHCs, we apply a lateral inhibition mechanism[17–19], to a region extending laterally to the pillar cell row. Lateral inhibition generates a disordered salt and pepper pattern in this region (Fig. 3a, left) and is further imposed throughout the simulations.

Given the observation of global shear motion in the OHC region (Fig. 2a–c) and the evidence for convergent extension[6,7,12], we hypothesized that global mechanical forces may play a role in the organization of HCs. Since these forces may not necessarily act at the apical surface, we also needed to consider the three-dimensional structure of the tissue. It has previously been shown that HCs lose their basal attachment and are free to migrate with respect to SCs that are still attached to the basement membrane[7,20]. Furthermore, after their differentiation, the nuclei of the HCs move away from the basement membrane and form a separate layer located at a sub-apical plane (schematic in Fig. 3b and Fig. S4b and images in Fig. S4c–e). The observation that HCs and SCs exhibit different lateral motion (Fig. 2c) and the spatial separation between their nuclei imply that the HCs and SCs are not subjected to the same forces.

We hence consider a two-layer model where global forces are exerted on the main cell bodies, namely on the HCs nuclei layer (lower layer in Fig. 3b and Fig. S4b) rather than on the apical layer (upper layer in Fig. 3b and Fig. S4b). Given the observed shear motion, we hypothesize that shear forces are exerted laterally on the nuclei of the HCs, possibly by the migrating Hensen cells touching them (lower layer in Fig. 3b and Fig. S4b and images in Fig. S4c–e'), and are directly conveyed to the apical layer. We, therefore, introduce the shear forces as external forces acting only on HCs rather than introducing shear on the apical boundary. Given that HCs maintain their lateral compaction with respect to the SCs (Fig. 2c), we also introduce lateral compression forces acting on HCs. We, therefore, model the total external forces at the apical layer, as diagonally oriented shear and compression forces acting on the OHCs towards the pillar cell row (left in Fig. 3c and Fig. S4a).

In addition to the global shear forces, we needed to take into account the observation that HCs rarely neighbor other HCs at the apical plane. To account for this behavior, we assume local repulsion forces between HCs (Fig. 3c, right). The repulsion forces may also arise from HCs touching each other at the HC nuclei layer (yellow arrows in Fig. 3b and Fig. S4b and images in

Fig. S4c–e'). Since the HCs in our movies seem to be rounder than SCs and exhibit less changes in their shapes, we further assumed in the model that the apical side of the HCs is less compressible and more contractile (tendency to maintain round shape) than that of SCs. Simulations performed after applying global shear and local repulsion forces to the 2D vertex model capture the main features of the patterning process of OHCs (Fig. 3d and Supplementary Video 8). More specifically, it captures the increase in hexagonal order of HCs and the decrease in the number of SC neighbors. We note that the global shear and local repulsion forces acting at the nuclei level and conveyed to the apical layer generate a situation analogous to shear-induced crystallization previously observed in granular and polymer physics[21–23].

A second aspect we wanted to capture in the model was the increase in the area of HCs relative to that of the SCs (Fig. 1g). Given the observed decrease in length of junctions between two supporting cells (SC:SC junctions) compared to the length of the junctions between HCs and SCs (HC:SC junctions), we introduced a second stage in our simulation in which we increased the SC:SC junction tension (Fig. 3e). At this stage we also introduced higher tension at the top boundary to constrict the OHC region. Applying these additional assumptions resulted in an almost-perfect confined checkerboard-like pattern (Fig. 3f. Supplementary Video 9).

Analysis of the order parameters in our simulated data showed that the number of SC neighbors decreases with simulation time and the hexagonal order parameter for OHCs increases with simulation time (Fig. 3g, h), similar to the experimental behavior (Fig. 1e, f). The simulations also capture the relative movements of the HCs and SCs. In the longitudinal direction shear forces on HCs drive the movement of HCs and SCs (that are dragged by HCs) where cells located more laterally move faster (Fig. S5a, left). In the lateral direction, the model captures the lateral movement of SCs with respect to the HCs mediated by the squeezing out processes observed experimentally (Fig. S5a, right). This process is the main contributor to the reduction in the number of SC neighbors. Although the second stage did not improve the order parameters, it was essential to obtain the observed final structure of the organ of Corti and capture the relative change in HC and SC area (Fig. 3i). We note that, for simplification purposes, the first and second stages are modeled sequentially although these processes may actually overlap.

The proposed model does fail to capture some aspects of the final pattern of the organ of Corti. In particular, the HCs in the model adopts rectangular shapes rather than the round shapes observed experimentally. This is due to the fact that in 2D vertex models, cell shapes are restricted to polygons with straight boundaries. Physically, HC rounding may occur due to the effect

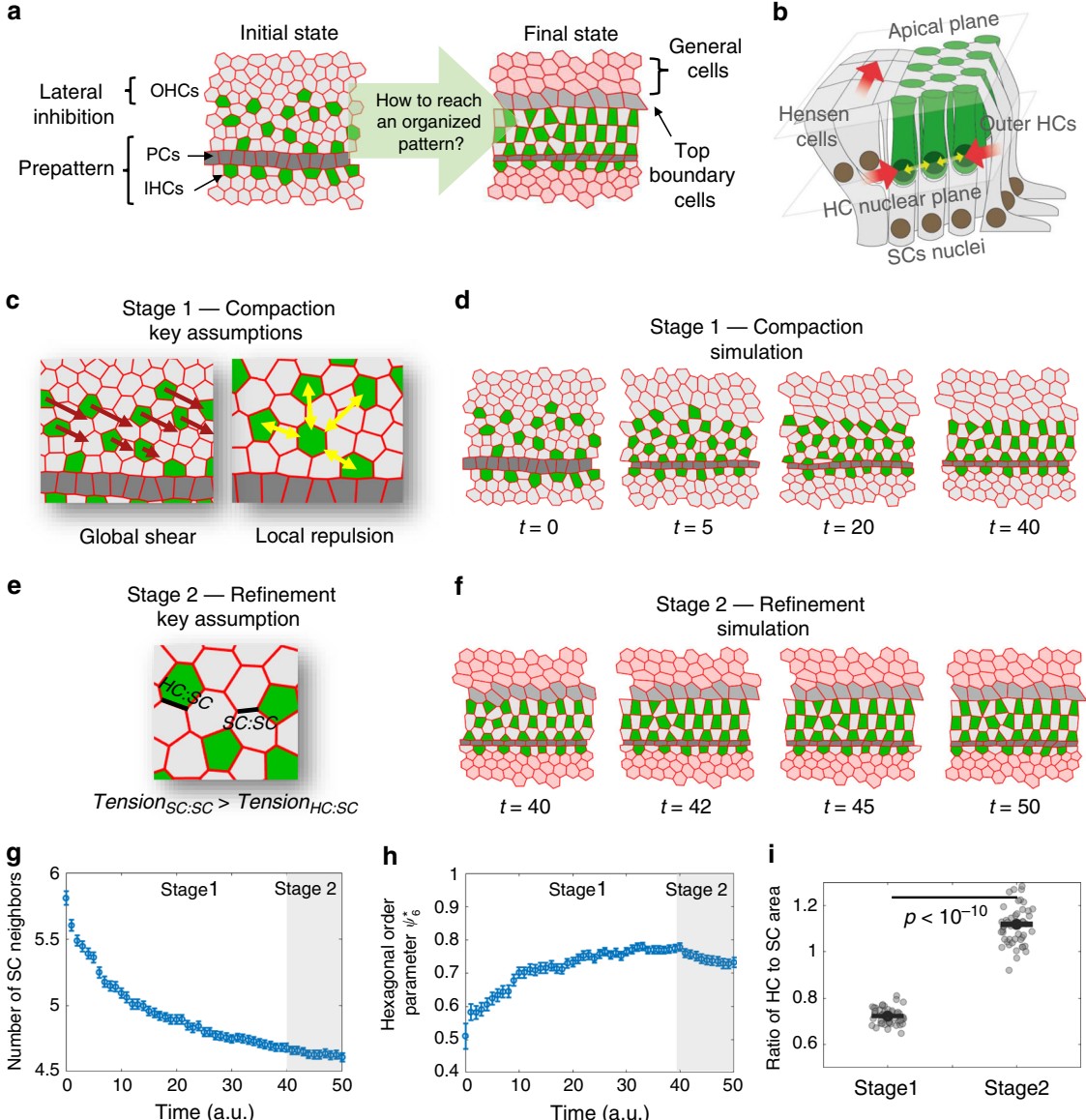

**Fig. 3 A mechanical model based on global shear and local repulsion explains patterning of the OHCs. a** Schematic of the initial state of the model and the desired final state. The initial state for the simulation begins with the IHCs and pillar cells already pre-formed. A lateral inhibition mechanism is used to define a disordered pattern of HCs and SCs at a certain distance from the pillar cells. **b** Schematic of the three-dimensional structure of the organ of Corti showing the forces assumed in the model (see also image in Fig. S3c–e). HCs nuclei apply steric repulsion at the HC nuclei plane (yellow arrows) while lateral shear motion driven by movement of Hensen cells leads to shear and compression on HCs (red arrows). **c** Schematic of the key assumptions for the first stage of the simulation (compaction stage). HCs are subjected to global shear forces (red arrows) and local repulsion forces (yellow arrows). **d** A filmstrip of the first stage in the simulation. Global shear and local repulsion forces lead to the formation of a compact state of HCs. Movie shown in Supplementary Video 8. **e** Schematic of the key assumption for the second stage (refinement stage). The tension in SC:SC junctions is increased relative to that of HC:SC junctions. **f** A filmstrip of the second stage in the simulation. This stage begins at the end of the compaction stage. Movie shown in Supplementary Video 9. **g**, **h** Simulations capture the dynamics of the order parameters observed experimentally. Similar to the analysis in Fig. 1e–f, the number of SC neighbors decreases (**g**) and the HC organization exhibits higher hexagonal order (**h**). **i** Simulations capture the change in HCs and SCs areas. Gray dots correspond to individual data points of the ratio of HC to SC areas at the end of stage 1 and stage 2, obtained from $n = 50$ simulations. Black dots represent average. Statistical analysis was done using two-sided two-sample $t$-test. Error bars in **g**–**i** indicate S.E.M. Full description of the simulations is provided in the methods. Parameters used are provided in Supplementary Table 1.

of higher internal pressure in HCs[24]. A more detailed model taking into account differences in internal pressures and bond curvatures will be required to capture these aspects of the tissue morphology. We also note that our model does not achieve perfect hexagonal patterning as some defects are still observed. Again, additional assumptions may be required for the final tuning of the pattern.

**Models based on local forces fail to capture HC patterning**. We also wanted to test if alternative models, which do not require global forces, can also capture the transition into ordered patterning of HCs. A potential mechanism that can lead to organized patterning is preferential adhesion of HCs to SCs over adhesion of SCs to SCs. This is equivalent to assuming only stage 2 occurs in our model. Such a model was previously used to describe

alternating patterning in the chick inner ear and has also been suggested to operate in the organ of Corti in mammals via interactions of Nectins[25,26]. To test whether such an alternative model captures the observed patterning features, we performed simulations where we assumed higher tension in SC:SC junctions compared to HC:SC junctions (Fig. S5b and Supplementary Video 10). The analysis showed that this class of models fails to capture several qualitative features observed experimentally: First, the hexagonal order in such simulations did not improve with simulation time leading to significantly more disordered patterns compared to simulations in our original model (Fig. S5c). Second, these models failed to capture both the shear motion and the lateral squeezing process observed experimentally (Fig. S5d). And finally, the reduction in the number of SC neighbors in these models relies mostly on delaminations of SCs and hence exhibit significantly higher number of delaminations compared to the original model (Fig. S5e). Simulations for both the original and the alternative models were repeated for a wide range of parameters. We found that changes by ±50% of most parameters in the original model had a relatively modest effect on the values of the order parameters in the simulations (Fig. S6a). Varying the parameters for the alternative models did not improve the values of the order parameters nor captured the main features observed experimentally (Fig. S6b).

**The apical side of SCs is more deformable than that of HCs.** As a means to verify the model we sought to generate predictions that can be tested experimentally by perturbing tissue mechanics. We first wanted to check how the tissue reorganizes upon a removal of a cell from the apical surface. The model predicted that since the apical side of SCs is more compressible than that of HCs, the apical closure following a cell removal would involve larger changes in the SC areas compared to those of HCs (Fig. 4a, b and Supplementary Video 11). To test this prediction, we have performed laser ablation experiments, where apical closure was tracked following the ablation of one or two cells in the organ of Corti, using a pulsed UV laser (Fig. 4c, d and Supplementary Video 12). Experiments were performed at E17.5 cochlear explants, where HCs and SCs can be easily distinguished based on their morphology. We found that following the ablation, the SCs surrounding the ablated area showed significantly larger change in their apical area compared to the surrounding HCs (Fig. 4d, e). We conclude that the apical surfaces of SCs are more deformable than those of HCs, allowing extra SCs to squeeze out as HCs get more compacted.

**Tension of SC:SC boundary is higher than that of HC:SC boundary.** We next focused on the effects of reducing the tensile forces in the tissue. Our model predicted that if the tension on all junctions was reduced and equated, the SC areas will increase relative to HC areas (Fig. 5a, b and Supplementary Video 13). To test this prediction experimentally, we added blebbistatin, a non-muscle myosin II (NMII) inhibitor, to E17.5 inner ear explants, 3 hours after the beginning of a time-lapse movie. It has been previously shown that adding blebbistatin to inner ear explants affects tension of SC:SC junctions and reduces movement of HCs[7,27]. As predicted by our model, the addition of blebbistatin indeed cause a significant increase in SC areas compared to HC areas (Fig. 5c, d and Supplementary Video 14). Control explants where blebbistatin was not added did not show significant changes in HC and SC areas (Fig. 5d). These results support the assumption that an increase in SC:SC tension contributes to the decrease in SC areas and the corresponding increase in HC areas.

## Discussion

The emergence of periodic patterns of cells during development is often associated with mechanisms based on regulatory circuits coordinated by intercellular signaling such as lateral inhibition or Turing patterning[17,18,28,29]. These processes, however, are inherently imprecise, as they depend on stochastic decisions at the single-cell level. Here we show that the transition from disordered to ordered pattern of OHCs in the organ of Corti is mainly driven by mechanical forces rather than signaling events. Our quantitative imaging analysis and mechanical modeling suggest that the formation of the ordered checkerboard-like pattern of OHCs and SCs is achieved by a combination of global shear and local repulsion forces between HCs and is refined by differential tension on SC:SC vs SC:HC junctions. This mechanism is analogous to the process of shear-induced crystallization, which have been shown to rapidly and efficiently drive initially disordered grains and molecules into crystalline order in multiple physical system[21–23].

Likely sources for the shear and repulsion forces are the active movement of Hensen cells, and the steric repulsion between the nuclei of HCs, respectively. The reduction in the number of SC neighbors may in principle occur due to (i) transdifferentiation of SCs to HCs, (ii) delaminations of SCs, and (iii) lateral squeezing out of SCs. Previous studies showed that transdifferentiation from SCs to HCs is a rare event[30] ruling it out as a major contributor to the process. While few delaminations are observed, our movie analysis shows that the squeezing out of SCs is the main process contributing to this reduction.

Interestingly, the observed shear motion can explain some global morphological properties of the cochlea. First, it has long been observed that the apical side of each SC is shifted towards the apex with respect to its main body but remains connected through a long diagonal protrusion[31,32] (schematic in Fig. S7). The observed shift can vary between positions along the cochlea, and between different mammalian species, ranging between 1 to 5 cells[32]. This observation can be explained by the shear motion as the apical sides of the SCs are dragged by the shearing HCs movement (Fig. 2c and Fig. S5a). Second, we note that, in addition to its role in HC organization, the observed shear motion may also underlie the spiral shape of the cochlea through asymmetric convergent extension. More generally, our proposed force-driven rearrangement mechanism may serve as a general principle for developmental patterning processes in other systems.

## Methods

**Mice.** The Math1-GFP line, J2XnGFP(Math1-nGFP tg) mice, were a gift from Jane E. Johnson, UT Southwestern[9] and maintained on a C57BL/ 6 background. *Rosa26-ZO1-EGFP* mice were obtained from RIKEN Laboratory[13] (accession no. CDB0260K) and maintained on a C57BL/6 background. All animal procedures were approved by the Animal Care and Use Committee at Tel Aviv University (04-16-014). Genotyping was performed using the KAPA HotStart Mouse Genotyping Kit (Sigma, KK7352) using GFP primers listed in Supplementary Table 2.

**Immunohistochemistry.** Mice were sacrificed by decapitation according to ethical guidelines and inner ears were dissected out in cold PBS and fixed in 4% paraformaldehyde (Electron Microscopy Sciences, cat: 15710) for 2 h at room temperature. For whole-mount imaging, sensory epithelia were exposed and separated from the inner ear. For cross sections, inner ears were processed in a Tissue Processor (Leica TP1020), positioned in paraffin blocks with a Histoembedder (Leica, Wetzlar, Germany) and sectioned using a microtome (Leica Jung RM2055). Paraffin serial sections (10 μm) were then dewaxed in xylene, rehydrated, and boiled for 3 min in unmasking solution (Vector Laboratories, cat: H-3301). Next, samples were incubated in 10% normal Donkey serum (Sigma, cat: D9663) with 0.2% Triton-X (Sigma, cat: T-8787) for 2 h at room temperature. Samples were then incubated with ZO-1 primary antibody diluted 1:250 (Thermo Fisher Scientific, cat: 339100) or MyoVIIa primary antibody diluted 1:250 (Proteus Biosciences, cat: 25-6790, lot:RC234446) overnight at 4 °C. Following three washes in PBS, samples were incubated with secondary antibodies of Cy™3 AffiniPure Goat Anti-Mouse IgG (H + L) (Jackson laboratories, cat: 115-165-062, lot:97726) or

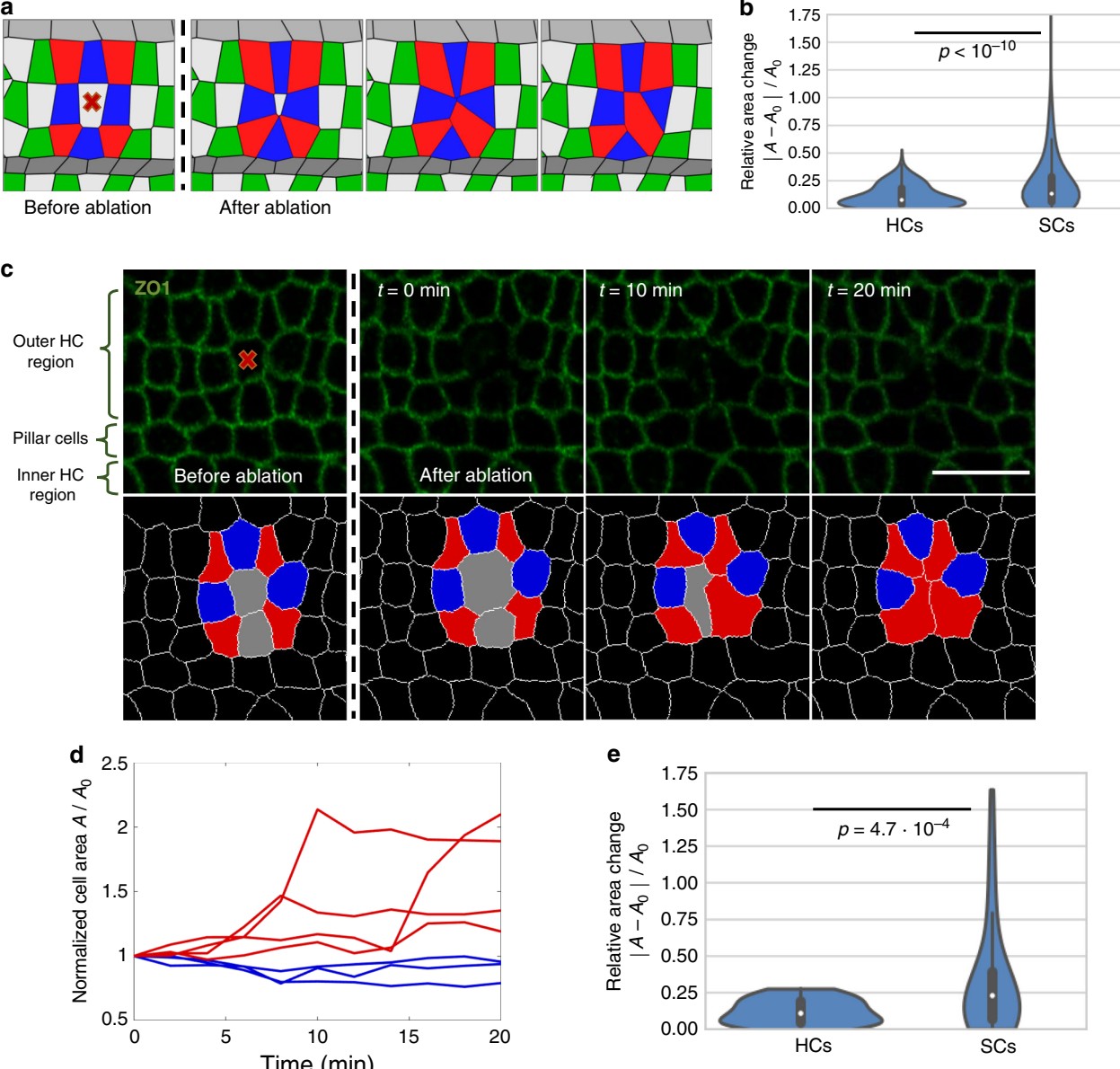

**Fig. 4 Laser ablation experiments show that SCs are more deformable than HCs. a** A filmstrip of a simulation modeling cell ablation (ablated cell marked with red "x"). Surrounding HCs (blue) and SCs (red) are marked. Movie shown in Supplementary Video 11. **b** Violin plot of the distributions of the relative change in area of HCs and SCs surrounding the ablated cells in the simulation. The presented data is taken from n = 825 simulations. **c** A filmstrip from a E17.5 ZO1-EGFP explant showing a laser ablation experiment. Ablation point is marked by red "x". Bottom row shows a segmented version of the filmstrip with ablated area (gray), nearest neighbors HCs (blue), and nearest neighbor SCs (red) marked. Movie shown in Supplementary Video 12. Scale bar: 10 μm. **d** Normalized area of the cells shown in **c**. The area of each cell is normalized by its initial area before the ablation. **e** Violin plot of the distributions of the relative change in area of HCs and SCs, 10 min after ablation. The presented data is taken from $n = 10$ experiments. In the violin plots in **b**, **e**, median is indicated by white point, interquartile range (IQR) is indicated by thick line, Q3 + IQR and Q1 − IQR are indicated by the edges of the thin line and kernel density estimate of the data is shown in blue. In both plots, the p-value from two-tailed Kolmogorov–Smirnov test is as indicated.

anti-rabbit Alexa Fluor 594 (Cell signaling, Cat: 8889, Lot: 7) for 2 h at room temperature. Stained samples were mounted on Cover glass 24x60mm thick. #1 (0.13-0.17 mm) slides (Bar-Naor Ltd. cat: BN1052441C) using a fluorescent mounting medium (GBI, cat: E18-18). Image acquisition was done with a ZEISS LSM 880 with Airyscan microscope (Zeiss).

**Organ of Corti explants**. The cochlea was dissected using sterile conditions under Nikon SM2 745 T Stereomicroscope (Nikon). Explant cultures were prepared from the cochleae of *ZO1-EGFP* transgenic mice. The cochleae were removed and placed in ice-cold PBS. Using two forceps the organ of Corti was gently freed from the capsule and separated from the stria vascularis. The tissue was oriented so that the apical surfaces of the hair cells were pointing down, directed toward the Matrigel Phenol Red Free solution (In vitro technologies, cat: FAL356237). Excess medium was removed and explanted tissue was allowed to attach to the Matrigel for

5–10 min in a 37 °C incubator with 5% $CO_2$ while avoiding drying of the tissue. After tissue attachment, Dulbecco's modified Eagle's medium (Biological industries, cat: 01-053-1 A) supplemented with N-2 Supplement (100X) (Thermo Fisher Scientific, cat: 17502001) and 1% FBS (Biological industries, cat: 04-007-1 A) was added gently. The plate was then placed in the 37 °C incubator of the microscope and as a control in the lab incubator. Cultures were kept for up to 48 h.

**Microscopy details**. Cochleae were imaged using Zeiss LSM 880 confocal microscope equipped with an Airyscan detector using 488 nm laser for GFP and 561 nm laser for Cy3. For fixed samples we used Plan-apochromat 63× oil-immersion objective with NA = 1.4. For live imaging we used C-apochromat 40× water-immersion objective with NA = 1.2. The microscope was equipped with a 37 °C temperature-controlled chamber and a $CO_2$ regulator providing 5% $CO_2$. The equipment was controlled using Zeiss software—"Zen black".

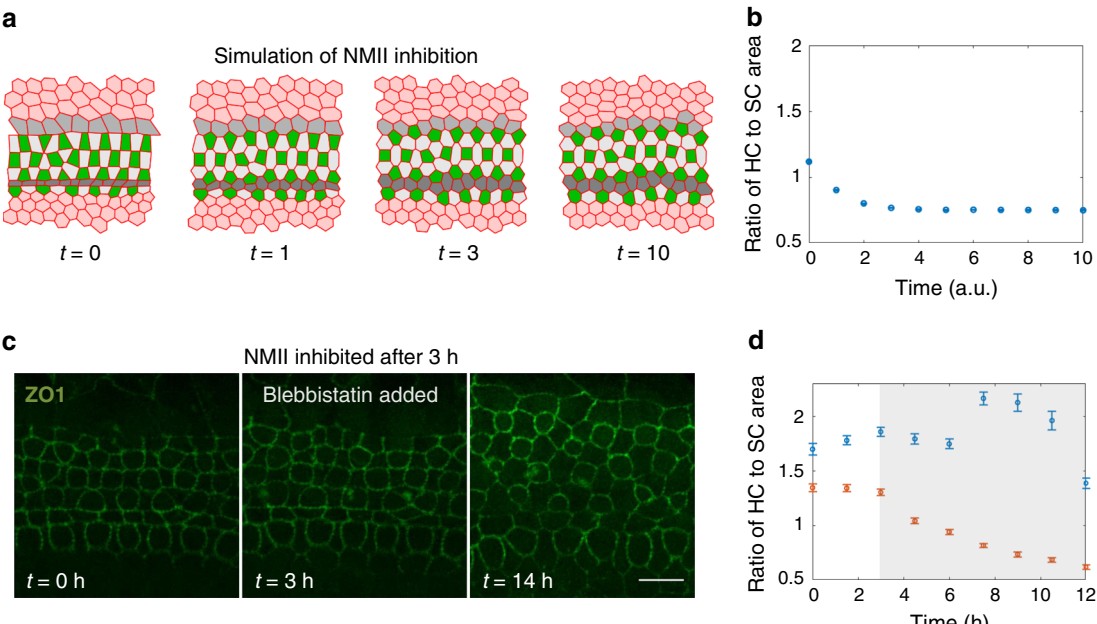

**Fig. 5 Model successfully predicts morphological changes induced by NMII inhibition. a** A filmstrip of a simulation modeling NMII inhibition. The addition of a NMII inhibitor, blebbistatin, was simulated by reducing and equating the tension for all junction types and overall contractility. Simulation starts at the end point of the simulations shown in Fig. 3. Movie shown in Supplementary Video 13. **b** Analysis of the ratio of HC area to SC area during the simulations. Model predicts a decrease in the ratio of HC to SC area. Data is averaged over $n = 30$ simulations. **c** A filmstrip from an E17.5 ZO1-EGFP explant treated with blebbistatin (10 μM). Blebbistatin was added 3 hours after the movie started. Movie shown in Supplementary Video 14. Scale bar: 10 μm. **d** Analysis of the ratio of HC area to SC area as a function of time demonstrated a decrease in the ratio of HC to SC areas in experiments where blebbistatin was added (red) compared to control experiments with no addition of blebbistatin (blue). Data is averaged over $n = 3$ experiments. Error bars in **b**, **d** indicate S.E.M.

**Laser ablation system**. Laser ablation experiments were performed using Rapp OptoElectronic UGA-42 Caliburn system equipped with a 355 nm UV short pulse laser. The ablation system was integrated to the microscope's hardware and software and controlled using Rapp software SysCon. In ablation experiments, cells were hit with 2000 short consecutive pulses over 2 s with a laser power of 2%.

**Image analysis**. Images of fixed samples were taken at high resolution (63× objective) and tiled together to form a full image of the cochlea. For fixed samples we used cochleae from transgenic mice expressing Math1-GFP, which is an early marker for HCs. Since the apical side of the organ of Corti is usually tilted relative to the focal plane, 15–30 z-stacks separated by 1 μm were taken at each position for both fixed and live imaging. The stacks were then projected to the focal plane using a maximum intensity projection method.

All data processing was performed off-line using custom built code in Matlab (MATLAB R2018b, the MathWorks Inc.). A semi-automatic analysis code was used for segmentation of the boundaries of the cells and for data extraction. In short, segmentation was done by applying filters to the image and using a watershed algorithm. Segmentation defects were corrected manually using a custom-made Matlab GUI. Different cell types for OHC ($OHC_1$ or $OHC_3$), $OHC_2$ or SC were manually marked relying on the Math1-GFP marker. In early stages of development, where the HC rows are not yet formed, we define $OHC_2$ as a HC that neither borders the pillar cells row nor the lateral border of the organ of Corti.

The initial segmentation output provides information about the identity (cell type), position (centroid coordinates), shape (area) and connectivity (number of neighbors, contact lengths) of each cell. We focus on analysis of the middle row of OHCs, $OHC_2$, since the HCs in this row only border SCs within the OHC region (i.e. in the final pattern they should have 4 SC neighbors). The morphological and order parameters presented in Fig. 1e–g are evaluated for each cell in $OHC_2$ using the segmentation output. These parameters are then grouped by the cochlear positions of the cells (see insert in Fig. 1e) and averaged. The morphological parameters and number of SC neighbors are directly extracted from the segmentation output. To get the ratio of HC to SC area, the area of each analysed HC is divided by the area of each of its adjacent SC neighbors (resulting in multiple values from each HC).

To calculate the hexagonal order of OHCs we use a modified version of the common $\psi_6$ order parameter[11]. Given a set of points in two-dimensional space, $\psi_6$ returns a local measure for hexagonal order. For each point $\vec{r}_j$ and its closest

neighbors $\{\vec{r}_k\}$, the parameter is given by:

$$\psi_6(j) = \frac{1}{N}\sum_{k=1}^{N} e^{6i\theta_{kj}} \tag{1}$$

where $N$ is the number of closest neighbors and $\theta_{kj}$ is the angle between $\vec{r}_{kj} = \vec{r}_k - \vec{r}_j$ and the x-axis. A set of points that sit on the vertices of a regular hexagon centered at $\vec{r}_j$ will give a $\psi_6$ value of one, while a randomly distributed set of points will give a lower value. $\psi_6$ cannot be used directly to estimate the level of hexagonal order of OHCs, since in the fully developed organ of Corti, neighboring HCs of $OHC_2$ cells are positioned on a regular hexagon stretched in one axis rather than a regular hexagon. Therefore, to get a correct hexagonal measure, the positions of the neighboring HCs need to be rescaled before using $\psi_6$. To do so we use arguments of simple geometry. The vertices of a regular hexagon sit on a circle. If the vertices are stretched by a factor $r_e$ in the direction of the main axis, the new vertices sit on an ellipse with a major to minor axis ratio of $r_e$. Therefore, to properly rescale the positions of the neighboring HCs, an ellipse is fitted to the centroids of the HCs and the ratio of major to minor axis $r_e^{fit}$ is acquired. Then, the centroid positions are squeezed by the factor $r_e^{fit}$ relative to the main axis direction and these modified positions are used to calculate $\psi_6$. The parameters calculated in this manner are marked as $\psi_6^*$. We note that although $\psi_6^*$ measures the fit to a stretched hexagon, we refer to it simply as hexagonal order parameter. The neighboring HCs of an $OHC_2$ cell are defined as cells with a shared junction in the Voronoi tessellation made by the centroids of all the HCs. The main axis of the hexagon is defined by fitting a line to the neighboring $OHC_2$ cells, including the analyzed cell.

**Cell tracking analysis**. To analyze the shear profile observed in movies along the base-to-apex axis we rotate the movies so that the IHCs are at the bottom of the field of view and the OHCs are at the top. In addition to relative motion between the cells, there is a global motion of the tissue. To consider only relative motion, we set the pillar cell row to be static with respect to the field of view, using image registration algorithm (Matlab—"imregdemons"). Then, cells were tracked manually using a semi-automated custom-made GUI. For each time point and each cell, the displacement from the initial position was calculated. The displacement was averaged separately for cells that were initially positioned at the medial half or lateral half of the OHC region. Cell type was determined by the cell

morphology at the final frame of the movie, where more convex cells were classified as HCs and more concave cells were classified as SCs.

**Intercalations analysis**. This analysis was done manually with the aid of an assisting GUI. The intercalations were identified visually and counted by clicking the location of intercalation. Each of these positions were marked for the duration of the intercalation to prevent repeated counts of the same transition. The intercalations were counted only for the OHC region as the IHC region is mostly organized at the developmental times we investigate. Finally, the total number of intercalations was divided by the number of cells in the OHC region and the duration of the movie.

**Cellular area movie analysis**. Movies of samples treated with blebbistatin and movies of cell ablation experiments were analyzed in the same way as fixed samples. The borders of the cells were segmented and the areas of the cells together with their identity were extracted from the segmentation output. For each analyzed frame in the movie the surface areas of HCs and SCs were calculated. For the blebbistatin experiments the average area of HCs was divided by the average area of SCs to get the ratio of HC to SC area. As a control, the same analysis was done for movies with no addition of blebbistatin. For the laser ablation experiments, the change in area of each cell was calculated 10 min after ablation for the neighboring cells of the ablated cells.

**Structure factor analysis**. To show the crystalline order in the organ of Corti we sought to calculate the structure factor of HCs at different developmental stages. Since the HCs are embedded in a spiral-shaped organ, the system must be straightened to align with the $x$-axis to get a meaningful structure factor. This process is done iteratively in the following manner. We start with the $\mathrm{OHC}_2$ cell closest to the base of the cochlea and set the origin at its position. To get an estimation for the local tilt angle of the cochlea, the average position of the neighboring HCs located farther to the apex is calculated, and the angle of this average position is measured relative to $x$-axis. Then, all HCs located farther to the apex relative to the cell at the origin are rotated in a negative angle to the one measured. Finally, the origin is set at the following $\mathrm{OHC}_2$ cell position (the next closest to the base) and the steps are repeated in an iterative manner.

Once aligned with the $x$-axis, the structure factor of the HCs is calculated as follows:

$$S(\vec{q}) = \frac{1}{N} \sum_{j=1}^{N} \sum_{k=i}^{N} e^{-i\vec{q}\cdot(\vec{r}_j - \vec{r}_k)} \tag{2}$$

For each developmental time, this was calculated for all the HCs in the base-mid region and summed for all repeats.

**Calculation of correlations and their _p_-value**. Statistical analysis shown in Figs. 2f, 3i, and S5e was performed using two-sample $t$-test. Statistical analysis shown in Fig. 4b, e was performed using two-sample Kolmogorov–Smirnov test. The number of samples is as indicated in each figure caption.

**Mechanical energy function for the 2D vertex model**. To describe the dynamics of the apical surface of the organ of Corti we use 2D vertex model previously used to describe morphological processes of epithelial tissues[15,16]. In this model each cell is described as a polygon which is uniquely defined by the position of its edges and vertices. A total mechanical energy associated with the position of the vertices, length of edges, and areas of the cells is minimized at each time step.

The energy function (also shown in Fig. S4a) is given by:

$$E = \sum_{n=1}^{N_c} \left\{ \frac{1}{2}\alpha_n \left(A_n - A_{n,0}\right)^2 + \sum_{ij_n} \gamma_n^{ij} l_n^{ij} + \frac{1}{2}\Gamma_n L_n^2 + \sum_{m=1}^{N_c} \sigma_{nm}\left(\frac{D_{nm}}{R_{nm}}\right)^{\kappa} \right\} \tag{3}$$

where $n$, $m$ are cell indices, $N_c$ is the total number of cells and $\langle ij \rangle_n$ are the pairs of adjacent vertices in cell $n$ (between vertices $i$ and $j$).

The first element in the energy function is an elastic term in the area of the cell $A_n$; each cell has a preferable area $A_{n,0}$ and each deviation from it results in a quadratic energy cost. Similar to a spring constant describing the rigidity of a spring, $\alpha_n$ describes the incompressibility of cell $n$. The second element is a linear term in the length of the junction $l_n^{ij}$. The parameter $\gamma_n^{ij}$ describes the tension of junction $ij$ in the sense that any increase in the junction's length results in an energy cost directly proportional to $\gamma_n^{ij}$. The third element is a quadratic term in the perimeter $L_n$ of cell $n$, and represents the susceptibility of the cells to deformations. $\Gamma_n$ describes the contractility of the cell, or the tendency of the cell to contract, and hence the tendency of the cell to round up (physically controlled by actomyosin cables in the cell cortex). Overall, we define the following cell types: HC, SC, pillar cell, top boundary cell or general cell outside the organ of Corti. This also allows setting different mechanical parameters for different cell types and junction types.

In addition to these standard terms in the energy function, we also define additional terms associated with global shear and local repulsion. In contrast to terms like tension and contractility that arise from interactions in the apical side of the tissue, we assume that global shear and local repulsion act on the main bodies

of the cells (in the nucleus region) which are in a sub-apical plane. Moreover, HCs nuclei and SCs nuclei are positioned on separate planes, and therefore are not exposed to the same forces. Since the main bodies of SCs, but not of HCs, are attached to the basal membrane, we assume that the shear forces effectively act on HCs and not SCs. Furthermore, given the close packing of cells at the HC nuclei layer, we assume that the local repulsion act on the HCs nuclei. While SCs might experience the same repulsion at the SC nuclei layer, this would not be conveyed to the apical surface through the thin non-rigid protrusions of the SCs. Overall, we consider a two-layer model where shear and repulsion forces originate at the HC nuclei layer and are then conveyed to HCs in the apical layer. While we do not explicitly calculate the forces in the HC nuclei layer, we assume that these result in "effective forces" acting on the HCs in the apical layer.

**Local repulsion between HCs**. The local repulsion is set in the energy function by the last term in Eq. (3). The steric hindrance between nuclei is modeled using a steep power law potential (that goes as $\sim 1/R_{nm}^{\kappa}$). The distance between the pair of cells $n$ and $m$, $R_{nm}$, is defined as the distance between the centers of masses of the cells. $\sigma_{nm}$, $D_{nm}$ and $\kappa$ quantifies the strength and behavior of these repulsion forces. Taking a high value for $\kappa$ results in a steeper potential with a decay radius close to $D_{nm}/2$ (with $\kappa \to \infty$ being a hard-sphere potential). We note that although we use a long-range power law term, the potential decays fast enough so that interaction between non-neighboring HCs is negligible relative to the other energy terms in Eq. (3).

Since the repulsion forces act only between HCs, we effectively have:

$$\sigma_{nm} = \begin{cases} \sigma & n, m \in \{HCs\}, n \neq m \\ 0 & \text{otherwise} \end{cases}, D_{nm} = \begin{cases} D & n, m \in \{HCs\}, n \neq m \\ 0 & \text{otherwise} \end{cases} \tag{4}$$

**Global shear forces in the model**. To model the shear force observed in experiments we decompose this force into two components of horizontal shear and vertical compression. Horizontal shear is assumed to have a linear gradient profile, vertical to the direction of motion, namely, if the motion is in the $x$ direction, the force changes with $y$. Hence, we can take this component to be proportional to the $y$ coordinate of each cell. For the vertical compression component we assume a quadratic energy term with respect to the $y$, namely, $E_n^{\mathrm{pull}} \propto \left(y_n^{\mathrm{CM}}\right)^2$ where $y_n^{\mathrm{CM}}$ is the $y$ coordinate of the center of mass of cell $n$ from the pillar cells. Hence, the effective compression force is proportional to the distance of the cell from the pillar cells. Overall, the global shear forces are taken into account by assuming external forces that acts on vertex $i$ of cell $n$ given by:

$$\left[\vec{F}_n^{\mathrm{ext}}\right]_i = \eta_n y_n^{\mathrm{CM}} \hat{x} + \zeta_n y_n^{\mathrm{CM}} \vec{\nabla}_i y_n^{\mathrm{CM}} \tag{5}$$

where $\eta_n$ and $\zeta_n$ describe the horizontal shear and vertical compression that act on HCs and $\vec{\nabla}_i$ is the divergence relative to vertex $i$ position. Since the shear and compression forces act only on HCs, we effectively have

$$\eta_n = \begin{cases} \eta & n \in \{HCs\} \\ 0 & \text{otherwise} \end{cases}, \zeta_n = \begin{cases} \zeta & n \in \{HCs\} \\ 0 & \text{otherwise} \end{cases} \tag{6}$$

The parameter values for different cell types and different junction types are given in Supplementary Table 1.

To account for random fluctuations of the vertices, we also add random noise for each of the vertices components as $\vec{F}^{\mathrm{ext}} \to \vec{F}^{\mathrm{ext}} + \vec{F}^{\mathrm{noise}}$ and randomize the components of $\vec{F}^{\mathrm{noise}}$ every few timesteps.

**Initial and boundary conditions**. To generate initial disordered 2D lattices, we use the method described in ref. [33]. In short, we begin with 12×12 hexagonal lattices and then run the 2D vertex model simulations, assigning random values for the tension ($\gamma_n^{ij}$) and preferable areas ($A_{n,0}$) every few time steps. For simplicity, we use periodic boundary conditions at the edges of the cell lattice.

As mentioned in the main text, we focus on the organization of the OHC region, so we begin the simulations in an initial state where the IHC and pillar cell rows already formed. The pillar cells form straight boundaries both with the IHC region and with the OHC region (Fig. 1c, d). In the simulation this is achieved by setting high tension ($\gamma_n^{ij}$) in junctions that separate pillar cells from non-pillar cells. Since the tension in the boundary junctions is higher, the model minimizes the boundary length to a straight line. In the rest of the simulation the pillar cell row serves as a boundary condition for the patterning of the outer HCs. To simulate the differentiation pattern of OHCs into HCs and SCs via lateral inhibition, we first define a region above the pillar cells in which lateral inhibition is active. We then apply the lateral inhibition process[17–19] by picking a random cell from this region and letting it differentiate into a hair cell only if it has no HC neighbors. This process is repeated until no additional hair cells can differentiate (according to the lateral inhibition rules). We note that we allow the differentiation of new HCs throughout the simulations according to the same rules, namely, if there is a SC within the OHC region that does not touch a HC, that cell can differentiate into a HC.

**Running the simulations**. The 2D vertex model simulates the dynamics of epithelial tissues by minimizing the net force on the system. Each term in the energy function and force expression can be calculated from the positions of all

the vertices (given a set of model parameters). In other words, the energy is a function of the position of all the vertices. For the purpose of convenience, the position of all the vertices is defined through a single vector structured such that entries $(2i − 1, 2i)$ of the vector represents the $(x, y)$ position of vertex $i$ respectively. If the total number of vertices is $N_v$, the position vector looks like this:

$$\vec{x} = \left( x^{(1)}, y^{(1)}, x^{(2)}, y^{(2)}, \ldots, x^{(N_v)}, y^{(N_v)} \right) \tag{7}$$

where $x^{(i)}, y^{(i)}$ are the $(X, Y)$ positions of vertex $i$. We assume that at each moment the system is trying to reach a steady state of zero net force on all of the vertices, meaning:

$$\left[ -\vec{\nabla} E + \vec{F}^{\text{ext}} \right]_{\vec{x}} = 0 \tag{8}$$

where $\vec{F}^{\text{ext}}$ is a vector of size $2N_v$ with the external force components of the vertices in the same format as $\vec{x}$ and the gradient is defined as:

$$\vec{\nabla} = \left( \frac{\partial}{\partial x^{(1)}}, \frac{\partial}{\partial y^{(1)}}, \frac{\partial}{\partial x^{(2)}}, \frac{\partial}{\partial y^{(2)}}, \ldots, \frac{\partial}{\partial x^{(N_v)}}, \frac{\partial}{\partial y^{(N_v)}} \right)$$

We can iteratively advance the vertices positions according to the gradient descent method in the following way:

$$\vec{x}_{t+1} = \vec{x}_t + \epsilon \left[ -\vec{\nabla} E + \vec{F}^{\text{ext}} \right]\big|_{\vec{x}_t} \tag{9}$$

where $t$ indicates the time step and $\epsilon$ is a small enough iteration parameter.

In parallel to advancing to the mechanical steady state, we introduce T1 and T2 transitions. A T1 transition initiates for junctions smaller than some threshold length. A T2 transition will initiate for each cell with an area smaller than some threshold area.

Once T1 transition has occurred, the initial transitioned junction is small and may go through a T1 process again. To prevent a situation where a certain junction goes through intercalations repeatedly at the same point, we allow a short relaxation time for the transitioned junction in which intercalations are not allowed.

Finally, to get exactly three rows of OHCs in the simulations, we limit the number of cells differentiating into OHCs so that the cells fit into exactly three rows (without extra HCs in the 4th row). We note, that this is necessary because of the imposed periodic boundary conditions in the simulations. Without the periodic boundary conditions, extra OHCs can be pushed forward towards the apex making sure an exact number of OHCs that fit three rows is reached.

All simulation codes were uploaded to public repository https://doi.org/10.5281/zenodo.4009613.

**Extracting statistics from the simulations**. For each time step in each simulation, statistics were taken in the same manner as in the analysis of fixed samples. The statistics include the number of SC neighbors, the hexagonal order parameter and the ratio of HC to SC area. Statistics are taken for HCs that don't touch the pillar cells or the top border. For Stage 1, we introduce the top border and only then take the statistics. For each time step, the statistics were averaged over all the simulations.

**Modeling blebbistatin addition, cell ablation, and alternative models**. In simulations modeling blebbistatin treatment, we begin at the last step of Stage 2 and then reduce and equate the tension by setting $\gamma_n^{ij} = \gamma_0$ for all junction types. In addition, shear and compression are suppressed by taking $\eta = \zeta = 0$. Parameters for these simulations are shown in Supplementary Table 1.

In simulations modeling cell ablation, we again begin at the last step of Stage 2. Then, we increase the contractility of a certain HC or SC to a very high value. This causes the cell to rapidly shrink and delaminate, effectively mimicking the behavior of an ablated cell. Except for the affected cell, the model parameters are the same as in Stage 2.

In the alternative model, the assumption is that the organization forms only due to tensile differences between different cellular junctions. More specifically, we set $\eta = \zeta = 0$ and there is no difference in compressibility and contractility between HCs and SCs ($\alpha_n = \alpha_0$, $\Gamma_n = \Gamma_0$ for all $n$). Parameters for these simulations are shown in Supplementary Table 1.

**Model parameters**. The parameter values used in the simulations are detailed in Supplementary Table 1.

## Data availability

Data supporting the findings of this paper are available from the corresponding author upon reasonable request. A reporting summary for this article is available as a Supplementary Information file. Source data are provided with this paper.

## Code availability

Matlab codes for lattice generation and running simulations were uploaded to the public repository https://doi.org/10.5281/zenodo.4009613.

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

## Acknowledgements
We would like to thank members of the Sprinzak and Avraham labs for their advice and comments on this work. We would like to acknowledge Assaf Zaritsky for advice on image analysis and Dmitri Rivkin for help with image analysis. We would like to thank Steve Blacklow, Michael Elowitz, Avigdor Eldar and Yasmine Meroz for fruitful discussions. Math1-GFP mice were a gift from Jane E. Johnson, UT Southwestern. This work has received funding from the European Research Council (ERC) under the European Union's Horizon 2020 research and innovation programme (Grant agreement No. 682161).

## Author contributions
This scientific study was conceived and planned by R.C., L.A-Z., and D.S. The inner ear explant experiments were performed by L.A-Z., R.C., S.W., O.L., and S.T. Support and expert advice for inner ear experiments were provided by K.B.A. The image analysis and image quantification were performed by R.C. and S.W., The ZO1-EGFP mouse and advice regarding its use were provided by F.M., The modeling was performed by R.C., M.H., S.B., and D.S. The paper was written by R.C., L.A-Z. and D.S.

## Competing interests
The authors declare no competing interests.
