## [Peer Review File · Nature Communications]

REVIEWER COMMENTS

Reviewer #3 (Remarks to the Author):

I was pleased to see the revised draft of this manuscript, "Shear-induced crystallization drives precise patterning of hair cells in the mammalian inner ear," transferred to Nature Communications. In my previous review, my main concerns had to do with using the language and concepts of crystal structure to describe the arrangements of cells the authors describe in their manuscript. The authors put in a lot of effort to improve the manuscript in this area, yet some major concerns remain, described below. Additionally, I was asked to evaluate whether the authors addressed the concerns raised by Referee #1. I cannot be certain about what that reviewer would think, but in my opinion the authors did an acceptable job at addressing the concerns. However, upon reading the comments from both of my fellow referees and looking carefully again at other aspects of the manuscript I previously did not scrutinize as much as other aspects, I have a few major concerns that align with those raised by other referees. My major concerns are described below.

1. I still have significant concerns about the analysis performed to show that the arrangement should be called a crystal, and based on the new analysis, I don't believe the authors should call their packing "hexagonal." The reason is that to compute ψ_6 , they need to stretch or squeeze their coordinates along one direction, which means that the right description could be any one of the other 2D lattices (monoclinic, orthorhombic, or square). Starting with any three of these, you can stretch or squeeze along one direction and end up with a hexagonal lattice. Thus, the HCs do not arrange into a hexagonal lattice. Using ψ_6 is fine – it shows that there is very good 6-fold coordination. But that does not mean a hexagonal lattice is formed. So, either the authors need to figure out what lattice is formed (which I don't think makes sense, based on the discussion below), or they stop calling it a hexagonal structure. Additionally, depending on how much stretching is done and along what direction, it may not even be true that the coordination between nearest neighbors is truly 6-fold. They should look at the other 2D lattices and see how one can draw hexagons on those too -- but we do not say those lattices have six-fold coordination between nearest neighbors. If they find that indeed the degree of stretching is small enough to conclude that they truly have six-fold coordination between nearest-neighbor HCs, they still should not use the term "hexagonal." Six-fold coordination is not the same as hexagonal symmetry.

2. Along the same lines as concern #1, I don't believe the structure factor analysis is interpreted or described correctly. First, the authors had to apply a deformation to the coordinate system before computing $S(q)$ – this already shows that what they have should not be called a crystal without some qualification. At best they should call it a deformed crystal. However, I am concerned that the term crystal still may not apply at all. In studying solid-state structures, there is a concept of "domain size" which is measured from the widths of Bragg peaks. A wide Bragg peak comes from a material with small crystalline domains, while a sharp Bragg peak comes from a material with large crystalline domains. As a reasonable estimate, one could just compute the reciprocal of the full width at half maximum (FWHM) of a Bragg peak, multiply it by two, and call that the domain size. The authors need to do this analysis to qualify the use of the term "crystal" when describing the HC arrangement. If they find the domain size is comparable to the size of their measurement footprint, then they can conclude that the system is crystalline over those length-scales, at least. However, if they find their domain size is the equivalent of just a couple of cells, then they should not call the HC arrangement crystalline. They can do this analysis along the two orthogonal axes, separately after performing the deformation transformation they currently do. The lateral-medial direction shows broader peaks because of the small number of rows along that axis. To correct for these finite-size effects, a reasonable approximation is to think of the measured Bragg peak as a convolution of the real peak with a finite-size peak, and treat the peaks as Gaussian. Thus, the convolution results in their widths being added in quadrature: $(\text{measured FWHM})^2 = (\text{real FWHM})^2 + (\text{finite-size FWHM})^2$. The finite-size FWHM is approximately the reciprocal of the system size along a given direction. Thus, it has a major blurring effect along the lateral-medial axis where there are only a few cell layers (as

seen in the $S(q)$ data in the revised draft), but it probably has negligible effects along the “long” axis. So, the real FWHM can be computed and the domain size will be two times its reciprocal.

3. There is a third level of analysis that should be done, also on the Bragg peaks. If they have a hexagonal lattice (which they do not appear to from need to do stretching and compute ψ^6 instead of ψ^6), their peaks should occur at predictable locations relative to one another: q_0 , $\sqrt{3}q_0$, $2q_0$, and so on. If not, they can analyze the relative peak locations along different directions and figure out what kind of lattice they do have (assuming we can call it a lattice).

4. Upon reading the revised draft, I became concerned about the authors’ strong claims that the changes in structure are mechanically driven. I’ve seen how hard it is for researchers at the interface of cell mechanics and developmental biology to prove that restructuring processes are driven by physical forces, and often the best one can do is to state that the observations are consistent with model predictions, without concluding that the mechanical mechanism has been shown. It doesn’t matter that one model qualitatively predicts the observations while other models don’t – there are, no doubt, more possibilities that would have to be ruled out to prove that this process is driven mechanically, in the absence of an experimental measurement of forces and responses. Thus, I believe the authors need to severely tone down statements about what they’ve shown (see my next concern, below).

5. There is too much hyperbole throughout the manuscript. For example, the authors say that “hair cells rearrange gradually into a precise checkerboard-like pattern.” I don’t think they’ve shown that the word “precise” should be used. Similarly, they say that their “findings suggest that mechanical forces drive precise hair cell patterning in a process strikingly analogous to the process of shear-induced crystallization.” The analogy is not striking – the evidence they provide is fairly weak. First, because they have not shown that crystallization occurs and second, because their evidence that the process is dominantly mechanical in nature and shear induced is indirect – there are no actual measurements of forces performed here. They also claim that they show that the cell rearrangements are “based on mechanical forces rather than signaling events.” They do not show this. They show that the observed rearrangements could be based on mechanical forces, based on conclusions drawn from their modeling, blebbistatin treatment, and ablation experiments. In my opinion, the authors are overstating what they’ve found. The authors also state that “each OHC from the middle HC row (OHC2) has exactly four SC neighbors,” which is clearly not true at any time-point during the transformation process, since there are finite-sized errorbars in the graphs describing the quantitative counts. The authors need to go through the entire manuscript and take away all the hyperbolic statements -- a good start would be to do word searches for terms like “precise” or “exact.”

6. The authors very briefly describe a “sensitivity” analysis of model parameters to check the robustness of their findings. They need to elaborate. As written, it is not clear what they did. Did they vary one parameter at a time? If so, what were the other parameters set to? Why not vary multiple parameters? Is it really justified to vary all parameters by $\pm 50\%$? Won’t the forces controlled by these parameters depend differently on these different parameters? Like if one term is proportional to the square of a particular parameter, then the effect of increasing by 50% would actually be the equivalent of increasing the whole term by 225%? As written, the whole process appears to be haphazard and arbitrary. Furthermore, sensitivity analysis is typically done to show that certain terms don’t matter, so they can be excluded from a model or a strategy for simplifying a model or combining terms can be determined. What I find puzzling is that the authors’ sensitivity analysis is used in almost the opposite way I would think of using it, attempting to show that their parameter values don’t matter. To me, this does not show robustness, it just says they haven’t found the dominating factors yet or varied their parameters correctly – right now it actually obscures the underlying physical mechanism more than it clarifies things.

7. The discussion about relative compressibility of cells is totally unjustified. First, just because the apical side of a cell expands or contracts, there is no reason to connect that to anything actually

compressing. The cell is a 3D object, and these cells are very long in the out-of-plane direction, if I understand things correctly. So any narrowing at the top could be accommodated by a tiny bit of bulging throughout the rest of the cell, with no net compression. If the authors are talking about "area compressibility" that's another thing. But, in that case, to show that material is actually compressing (concentrating) in a plane, a lot more needs to be done. Is the membrane compressing? Cortical actin? Membrane proteins and receptors? What is actually compressing? It seems to me that they're just looking at cell-cell interfaces move and calling that compression. Moreover, the term "compressibility" is an equilibrium concept that may not apply here. These cells are no doubt generating different levels of tension actively, and the heterogeneous area changes the authors observe could be just reflections of that tension driving collective structural changes following disruption (ablation or blebbistatin, for example). These processes can occur without any differences in area compressibility between cells. Unless the authors can contrive a way to apply a known tension to the borders of cells and measure their resulting area changes, they should not claim that what they're observing has anything to do with compressibility. By contrast, they should just state what they're observing – different cells exhibit different levels of expansion and contraction in response to perturbations.

In summary, I think the authors need much more precision in their language, a more care in their analysis, and they need to tone down their claims throughout the manuscript. Finally, considering all these concerns, I think they need to change their title. I support the idea of keeping "shear induced crystallization" in the title, but they need to be clear that what they have is an analogy, not a true example. Perhaps something like "Comparison of patterning of hair cells in the mammalian inner ear and shear-induced crystallization."

A point-by-point response to the reviewer comments

I was pleased to see the revised draft of this manuscript, "Shear-induced crystallization drives precise patterning of hair cells in the mammalian inner ear," transferred to Nature Communications. In my previous review, my main concerns had to do with using the language and concepts of crystal structure to describe the arrangements of cells the authors describe in their manuscript. The authors put in a lot of effort to improve the manuscript in this area, yet some major concerns remain, described below. Additionally, I was asked to evaluate whether the authors addressed the concerns raised by Referee #1. I cannot be certain about what that reviewer would think, but in my opinion the authors did an acceptable job at addressing the concerns. However, upon reading the comments from both of my fellow referees and looking carefully again at other aspects of the manuscript I previously did not scrutinize as much as other aspects, I have a few major concerns that align with those raised by other referees. My major concerns are described below.

1. I still have significant concerns about the analysis performed to show that the arrangement should be called a crystal, and based on the new analysis, I don't believe the authors should call their packing "hexagonal." The reason is that to compute ψ_6^* , they need to stretch or squeeze their coordinates along one direction, which means that the right description could be any one of the other 2D lattices (monoclinic, orthorhombic, or square). Starting with any three of these, you can stretch or squeeze along one direction and end up with a hexagonal lattice. Thus, the HCs do not arrange into a hexagonal lattice. Using ψ_6^* is fine – it shows that there is very good 6-fold coordination. But that does not mean a hexagonal lattice is formed. So, either the authors need to figure out what lattice is formed (which I don't think makes sense, based on the discussion below), or they stop calling it a hexagonal structure. Additionally, depending on how much stretching is done and along what direction, it may not even be true that the coordination between nearest neighbors is truly 6-fold. They should look at the other 2D lattices and see how one can draw hexagons on those too -- but we do not say those lattices have six-fold coordination between nearest neighbors. If they find that indeed the degree of stretching is small enough to conclude that they truly have six-fold coordination between nearest-neighbor HCs, they still should not use the term "hexagonal." Six-fold coordination is not the same as hexagonal symmetry.

We thank the reviewer for raising this point. We agree that the underlying crystal formed by the HCs is not necessarily a hexagonal lattice according to solid state definitions. In fact, we estimate that it is a monoclinic crystal as the two basis vectors have an angle of approximately 55 degrees, rather than 60 degrees in a hexagonal lattice. We therefore no longer use the term 'hexagonal lattice' in our manuscript (it appeared once in the previous manuscript).

However, we do think that the terms hexagonal order and hexagonal patterning correctly describes the symmetry observed. We clearly define what we mean by hexagonal order through the definition of stretched hexagonal order parameter ψ_6^* . We note that as the system becomes more organized the correction for stretching becomes rather small (up to a factor of 1.2 between the axes at P0).

2. Along the same lines as concern #1, I don't believe the structure factor analysis is interpreted or described correctly. First, the authors had to apply a deformation to the coordinate system before computing $S(q)$ – this already shows that what they have should not be called a crystal without some qualification. At best they should call it a deformed crystal. However, I am concerned that the term crystal still may not apply at all. In studying solid-state structures, there is a concept of "domain size"

which is measured from the widths of Bragg peaks. A wide Bragg peak comes from a material with small crystalline domains, while a sharp Bragg peak comes from a material with large crystalline domains. As a reasonable estimate, one could just compute the reciprocal of the full width at half maximum (FWHM) of a Bragg peak, multiply it by two, and call that the domain size. The authors need to do this analysis to qualify the use of the term “crystal” when describing the HC arrangement. If they find the domain size is comparable to the size of their measurement footprint, then they can conclude that the system is crystalline over those length-scales, at least. However, if they find their domain size is the equivalent of just a couple of cells, then they should not call the HC arrangement crystalline. They can do this analysis along the two orthogonal axes, separately after performing the deformation transformation they currently do. The lateral-medial direction shows broader peaks because of the small number of rows along that axis. To correct for these finite-size effects, a reasonable approximation is to think of the measured Bragg peak as a convolution of the real peak with a finite-size peak, and treat the peaks as Gaussian. Thus, the convolution results in their widths being added in quadrature: $(\text{measured FWHM})^2 = (\text{real FWHM})^2 + (\text{finite-size FWHM})^2$. The finite-size FWHM is approximately the reciprocal of the system size along a given direction. Thus, it has a major blurring effect along the lateral-medial axis where there are only a few cell layers (as seen in the $S(q)$ data in the revised draft), but it probably has negligible effects along the “long” axis. So, the real FWHM can be computed and the domain size will be two times its reciprocal.

We agree that the crystal is indeed a deformed crystal as the HCs are arranged along the spiral-shaped cochlea. Each structure factor that is presented in the paper is a combination of structure factors from different samples with some biological variability. The purpose of this is to present in a compact figure the average crystallization process throughout time, but domain sizes should not be calculated from these due to the biological variability. We have now calculated the structure factor for a single completely developed cochlea at P8 (see Fig. R1 – new Fig. S1c-d in the manuscript). The structure factor, taken from a region of a P8 cochlea, shows clear crystalline order. It exhibits multiple peaks corresponding to higher order spatial frequencies and the inverse of the width of the main Bragg peak in the x-axis is of the order of the domain analyzed in this image ($>100\mu\text{m}$). The width in the y-axis indeed fits well the width expected from 3 rows. Overall, it is clear from the analysis that long range crystalline order of HCs is established.

We have now added the following sentence to the manuscript: ‘In a fully ordered organ of Corti the HCs are organized in a deformed crystal structure along the spiral shaped cochlea’.

Figure R1. (Fig. S1c-d in the manuscript) Crystalline order of OHCs. (a) An image of a mid region from a fully developed cochlea at P8, stained with MyoVIIa (marker for HCs) and taken at the apical side of the tissue (top). Scatter plots show the centroids of the OHCs apical cross section before (middle) and after (bottom) applying straightening transformation (see methods). Scale bar: $20\mu\text{m}$. (b) Structure factor of the transformed OHCs centroids in (a). The observed Bragg peaks demonstrates the high level of crystalline order in the system. Image in inset shows a closeup of the center peak with its adjacent sub peaks, as expected from a finite size system.

3. There is a third level of analysis that should be done, also on the Bragg peaks. If they have a hexagonal lattice (which they do not appear to from need to do stretching and compute ψ^6 instead of ψ^6), their peaks should occur at predictable locations relative to one another: q_0 , $\sqrt{3}q_0$, $2q_0$, and so on. If not, they can analyze the relative peak locations along different directions and figure out what kind of lattice they do have (assuming we can call it a lattice).

As mentioned, this is indeed not a hexagonal crystal but rather a monoclinic one. We do not think that the characterization of the crystalline symmetry is in the main focus of this manuscript and hence do not expand on it. However, it may be an interesting topic to expand in future work.

4. Upon reading the revised draft, I became concerned about the authors' strong claims that the changes in structure are mechanically driven. I've seen how hard it is for researchers at the interface of cell mechanics and developmental biology to prove that restructuring processes are driven by physical forces, and often the best one can do is to state that the observations are consistent with model predictions, without concluding that the mechanical mechanism has been shown. It doesn't matter that one model qualitatively predicts the observations while other models don't – there are, no doubt, more possibilities that would have to be ruled out to prove that this process is driven mechanically, in the absence of an experimental measurement of forces and responses. Thus, I believe the authors need to severely tone down statements about what they've shown (see my next concern, below).

Changes in structure can be driven by mechanical forces or by changes in cellular biochemistry possibly induced by cellular signaling. We did not claim that the organization is driven only by mechanical forces, but rather that it is mainly driven by it. We also mention that the tension in cellular junctions is a key aspect in the refinement of the structure, which relates to the chemical adherence between the cells.

5. There is too much hyperbole throughout the manuscript. For example, the authors say that "hair cells rearrange gradually into a precise checkerboard-like pattern." I don't think they've shown that the word "precise" should be used. Similarly, they say that their "findings suggest that mechanical forces drive precise hair cell patterning in a process strikingly analogous to the process of shear-induced crystallization." The analogy is not striking – the evidence they provide is fairly weak. First, because they have not shown that crystallization occurs and second, because their evidence that the process is dominantly mechanical in nature and shear induced is indirect – there are no actual measurements of forces performed here. They also claim that they show that the cell rearrangements are "based on mechanical forces rather than signaling events." They do not show this. They show that the observed rearrangements could be based on mechanical forces, based on conclusions drawn from their modeling, blebbistatin treatment, and ablation experiments. In my opinion, the authors are overstating what they've found. The authors also state that "each OHC from the middle HC row (OHC2) has exactly four SC neighbors," which is clearly not true at any time-point during the transformation process, since there are finite-sized errorbars in the graphs describing the

quantitative counts. The authors need to go through the entire manuscript and take away all the hyperbolic statements -- a good start would be to do word searches for terms like "precise" or "exact."

We appreciate the need to use accurate terms, and we have modified some of the terms according to the suggestions. However, we feel that the term 'precise' should stay in place. The following specifies our response to each term used.

Regarding the term 'precise patterning' – We believe the term fits well as a description of the transition that the organ of Corti undergoes. By definition 'Precision' refers to the closeness of two or more measurements to each other. Thus, by 'precise patterning' we mean that the HCs and SCs gets close to a checkerboard like pattern. We show that the number of SC neighbors gets close to 4 and that the HCs arrange in a pattern close to hexagonal pattern.

Regarding the term 'strikingly analogous' – we think it's a matter of taste or style rather than a matter of definition. We do find that the analogy is striking (given we have now shown the crystalline order). We will be OK with changing it to highly analogous.

Regarding 'based on mechanical forces' – We have now changed the sentence to 'checkerboard-like pattern of hair cells and supporting cells in the mammalian hearing organ, the organ of Corti, is likely based on mechanical forces rather than signaling events'

Regarding the term 'exactly' – We have now changed the sentence to 'each OHC from the middle HC row (OHC2) has almost always four SC neighbors'

6. The authors very briefly describe a "sensitivity" analysis of model parameters to check the robustness of their findings. They need to elaborate. As written, it is not clear what they did. Did they vary one parameter at a time? If so, what were the other parameters set to? Why not vary multiple parameters? Is it really justified to vary all parameters by +/-50%? Won't the forces controlled by these parameters depend differently on these different parameters? Like if one term is proportional to the square of a particular parameter, then the effect of increasing by 50% would actually be the equivalent of increasing the whole term by 225%? As written, the whole process appears to be haphazard and arbitrary. Furthermore, sensitivity analysis is typically done to show that certain terms don't matter, so they can be excluded from a model or a strategy for simplifying a model or combining terms can be determined. What I find puzzling is that the authors' sensitivity analysis is used in almost the opposite way I would think of using it, attempting to show that their parameter values don't matter. To me, this does not show robustness, it just says they haven't found the dominating factors yet or varied their parameters correctly – right now it actually obscures the underlying physical mechanism more than it clarifies things.

The sensitivity analysis is a standard procedure performed in many mathematical models in biology. The aim of this procedure is to show that the model is not sensitive to the specific set of parameters used. This analysis checks that the model is not 'fine tuned' to a very narrow parameter range and fails when varying the parameters just by a bit. A 'sensitive model' is unlikely to be employed in biological systems that often are very noisy. This is often referred to as robustness of a model. (Barkai and Shilo, Mol. Cell 2007, <https://doi.org/10.1016/j.molcel.2007.11.013>)

More specifically, we varied one parameter at a time by 50% – this is now explicitly mentioned in the caption of supplementary Figure S6. The other parameters were set to the values used in the main simulation (specified in table S1). The reason we do not perform a multi-dimensional parameter search is that we did not intend to find the “global minimum” for the system. We think this will not add significantly to the claims of the paper. The choice of 50% is indeed arbitrary but 20-50% change is common in these type of sensitivity analysis.

We have also removed the word ‘robustness’ from the caption of Fig. S6.

7. The discussion about relative compressibility of cells is totally unjustified. First, just because the apical side of a cell expands or contracts, there is no reason to connect that to anything actually compressing. The cell is a 3D object, and these cells are very long in the out-of-plane direction, if I understand things correctly. So any narrowing at the top could be accommodated by a tiny bit of bulging throughout the rest of the cell, with no net compression. If the authors are talking about “area compressibility” that’s another thing. But, in that case, to show that material is actually compressing (concentrating) in a plane, a lot more needs to be done. Is the membrane compressing? Cortical actin? Membrane proteins and receptors? What is actually compressing? It seems to me that they’re just looking at cell-cell interfaces move and calling that compression. Moreover, the term “compressibility” is an equilibrium concept that may not apply here. These cells are no doubt generating different levels of tension actively, and the heterogeneous area changes the authors observe could be just reflections of that tension driving collective structural changes following disruption (ablation or blebbistatin, for example). These processes can occur without any differences in area compressibility between cells. Unless the authors can contrive a way to apply a known tension to the borders of cells and measure their resulting area changes, they should not claim that what they’re observing has anything to do with compressibility. By contrast, they should just state what they’re observing – different cells exhibit different levels of expansion and contraction in response to perturbations.

To more accurately describe the change in apical areas of HCs and SCs following laser ablation, we now use the term ‘deformable’ (the apical area of SCs are more deformable than that of HCs).

In the model, the parameter describing the dependence of the mechanical energy on area (α_n) is by definition area compressibility. We hence use the term compressibility whenever we refer to the model in the text (dropping the ‘area’ for abbreviation since the model is in 2D).

We do note that the apical side of the cells (the cuticular plate) contains dense actin mesh, and we believe that the regulation of the apical area of the cells is controlled by active processes in this region. Hence, it is likely that area compressibility is indeed the relevant physical quantity for this system. This is however well beyond the current scope of the manuscript.

In summary, I think the authors need much more precision in their language, a more care in their analysis, and they need to tone down their claims throughout the manuscript. Finally, considering all these concerns, I think they need to change their title. I support the idea of keeping “shear induced crystallization” in the title, but they need to be clear that what they have is an analogy, not a true example. Perhaps something like “Comparison of patterning of hair cells in the mammalian inner ear and shear-induced crystallization.”

We believe the manuscript brings strong evidence for the role of shear (this is the first observation of shear motion in this system) and the involvement of cell mechanics in the system. Given all our observations and analysis, we think that the current title accurately captures the main conclusions.

REVIEWER COMMENTS

Reviewer #3 (Remarks to the Author):

I appreciate the authors addressing most of my concerns. Unfortunately, they still use the terminology of crystal structure incorrectly and until they describe their findings with accuracy, I cannot recommend publication. This concern and a few major issues are described below.

1. The authors should not use the term "long range order" to describe the structures they observe. "Long range order" has a well established technical meaning. To demonstrate "long range order" the authors would have to show that the widths of their Bragg-peaks are resolution limited in reciprocal space (or limited by the system size in real-space). From their new analysis, it appears that the crystalline domain size is on the order of 13 cell spacings (just over 100 microns). This is much smaller than the size of the image analyzed in figure S1c and dramatically smaller than the entire organ of Corti. Since the focus of this manuscript is not on the details of crystal structure, the right thing to do is to accurately describe what they see: crystalline order with characteristic domain sizes of around 13 cell-spacings. I think even with the information they've gathered so far, they could legitimately call it "quasi-long-range order" which is a technical term used to describe order in liquid crystals.

2. In response to one of my previous concerns, the authors state, "this is indeed not a hexagonal crystal but rather a monoclinic one. We do not think that the characterization of the crystalline symmetry is in the main focus of this manuscript and hence do not expand on it." This is fine, except the authors still focus on hexagonal order in the manuscript. See this excerpt:

"The second order parameter defines a measure for the hexagonal packing of HCs. In a fully ordered organ of Corti the HCs are organized in a deformed crystal structure along the spiral shaped cochlea (Fig. S1c-d). Each HC from OHC2 is at the center of a regular hexagon stretched in one axis, formed by its closest HC neighbors (Fig. 1d, bottom row). To obtain a measure of the local hexagonal order, we use a modified version of the common hexagonal order parameter ψ_6 . This parameter value is one for a perfect regular hexagon and close to zero for an uncorrelated set of points. To measure the hexagonal order of HCs, we used a modified order parameter, termed ψ_6^* , which takes into account the stretching of the pattern along the base-to-apex axis. ψ_6^* was calculated for the centroids of neighboring HCs of each cell from OHC2 by first estimating the degree of stretching and then calculating for the scaled centroid positions (see methods). Analysis across all the cochleae measured, showed that the hexagonal order parameter, ψ_6^* , increases in value with the developmental stage, indicating the gradual organization of the HCs into a hexagonal pattern."

Based on their reply to me, this excerpt from the manuscript is misleading. Why not just say that the cells arrange into a monoclinic lattice, but that to facilitate measuring you choose to compute ψ_6^* ? This would be fine. However, in this case, it is not true to state that the cells have 6 nearest neighbors -- in a monoclinic lattice each object has four nearest neighbors. By stretching or squeezing a monoclinic lattice into a hexagonal lattice, you get 6. I think with only a couple of additional sentences the authors can describe things correctly. They really must go through the entire manuscript, however, and make sure not to give the impression that the crystal has hexagonal symmetry, or that the cells end up with six nearest neighbors. Currently, the description is misleading.

3. I still take issue with the liberal use of the word "precise" throughout the manuscript. In their reply to me, the authors state that "by 'precise patterning' we mean that the HCs and SCs gets close to a checkerboard like pattern. We show that the number of SC neighbors gets close to 4 and that the HCs arrange in a pattern close to hexagonal pattern." My response is: how is it reasonable to say "close to" is the same as "precise"? To avoid the semantic argument, you should say *how* precise if you want call something "precise." If the authors want to expend the effort quantifying their precision, then I

encourage them to do so. Otherwise, from my read of the manuscript, they should not use the term so frequently or loosely.

4. I appreciate the authors explaining their use of the term "compressibility." I am fine with them dropping "area" from "area compressibility." Even in light of this explanation, though, some of their description is still misleading. For example, the sentence that starts with, "The model predicted that since SCs are more compressible than HCs... ." From their explanation, SCs are NOT more compressible than HCs -- their apical areas are more compressible. The authors need to clean up this description to better reflect what they mean.

5. The manuscript title is still an overstatement. The authors have not shown that what they are observing is shear-induced crystallization. They have shown that the patterning process is analogous to shear-induced crystallization. To show that the process is shear-induced crystallization, the authors would have to either apply a controlled shear stress by external means, or figure out how to measure the hypothesized cell-generated shear stress, then link it to the crystallization process. I understand this may be difficult or maybe impossible, but the indirect evidence shown by the authors is not sufficient -- and the other referees expressed the same concerns. And once again, I object to the use of the word "precise" without qualification, given what the authors have actually shown.

I hope the authors appreciate that my concerns can be addressed with small amounts of editing. I will be able to recommend publication once the manuscript is free of inaccurate terminology and overstatements of what has been found.

Point-by-point response to review #3

Response is in blue font.

I appreciate the authors addressing most of my concerns. Unfortunately, they still use the terminology of crystal structure incorrectly and until they describe their findings with accuracy, I cannot recommend publication. This concern and a few major issues are described below.

We thank the reviewer for the comments.

1. The authors should not use the term "long range order" to describe the structures they observe. "Long range order" has a well established technical meaning. To demonstrate "long range order" the authors would have to show that the widths of their Bragg-peaks are resolution limited in reciprocal space (or limited by the system size in real-space). From their new analysis, it appears that the crystalline domain size is on the order of 13 cell spacings (just over 100 microns). This is much smaller than the size of the image analyzed in figure S1c and dramatically smaller than the entire organ of Corti. Since the focus of this manuscript is not on the details of crystal structure, the right thing to do is to accurately describe what they see: crystalline order with characteristic domain sizes of around 13 cell-spacings. I think even with the information they've gathered so far, they could legitimately call it "quasi-long-range order" which is a technical term used to describe order in liquid crystals.

The domain size we previously mentioned (100 microns) is a rough estimate, calculated as 2 over the FWHM of the structure factor. Given that the analyzed sample length is about 300 microns (Fig. S1c in the manuscript), this shows that the domain size is in the same order of magnitude. Taking a more accurate approach, we analytically calculate the structure factor for a finite hexagonal lattice (as a convenient private case of a monoclinic lattice, where the corrections for monoclinic should be negligible) (see appendix). The more accurate domain size, derived by measuring the position of the structure factor's first minimum, is 299 microns, similar to the analyzed sample size. Hence, the domain size is at least ~ 40 cell spacings which seems to justify the use of 'long range order' even according to this strict definition. We note, that for biological systems this range is certainly considered as 'long range order' and hence we retain the term.

2. In response to one of my previous concerns, the authors state, "this is indeed not a hexagonal crystal but rather a monoclinic one. We do not think that the characterization of the crystalline symmetry is in the main focus of this manuscript and hence do not expand on it." This is fine, except the authors still focus on hexagonal order in the manuscript. See this excerpt:

"The second order parameter defines a measure for the hexagonal packing of HCs. In a fully ordered organ of Corti the HCs are organized in a deformed crystal structure along the spiral shaped cochlea (Fig. S1c-d). Each HC from OHC2 is at the center of a regular hexagon stretched in one axis, formed by its closest HC neighbors (Fig. 1d, bottom row). To obtain a measure of the local hexagonal order, we use a modified version of the common hexagonal order parameter ψ_6 . This parameter value is one for a perfect regular hexagon and close to zero for an uncorrelated set of points. To measure the hexagonal order of HCs, we used a modified order parameter, termed ψ_6^* , which takes into account the stretching of the pattern along the base-to-apex axis. ψ_6^* was calculated for the centroids of

neighboring HCs of each cell from OHC2 by first estimating the degree of stretching and then calculating for the scaled centroid positions (see methods). Analysis across all the cochleae measured, showed that the hexagonal order parameter, ψ_6^* , increases in value with the developmental stage, indicating the gradual organization of the HCs into a hexagonal pattern."

Based on their reply to me, this excerpt from the manuscript is misleading. Why not just say that the cells arrange into a monoclinic lattice, but that to facilitate measuring you choose to compute ψ_6^* ? This would be fine. However, in this case, it is not true to state that the cells have 6 nearest neighbors -- in a monoclinic lattice each object has four nearest neighbors. By stretching or squeezing a monoclinic lattice into a hexagonal lattice, you get 6. I think with only a couple of additional sentences the authors can describe things correctly. They really must go through the entire manuscript, however, and make sure not to give the impression that the crystal has hexagonal symmetry, or that the cells end up with six nearest neighbors. Currently, the description is misleading.

As the reviewer suggested, we have now explicitly mentioned in the manuscript that the HCs organize into a monoclinic deformed crystal. Moreover, we have now made it clear in the text that what we refer to as "higher hexagonal order" is just higher values of ψ_6^* , which simply measures how neighboring cells are close to a stretched hexagon (and not a measure of regular hexagonal symmetry). We have omitted all references to hexagonal crystals and only refer to hexagonal order as described here.

Regarding the number of neighbors, since during development the HCs structure is not yet a crystal, we do not use the crystallographic definition of nearest neighbors but rather define neighboring points as ones that share a cellular junction in the Voronoi tessellation (described in the methods). Therefore, using our definition, a cell in a monoclinic lattice (that is close to a hexagonal one) will have 6 neighbors (see figure below).

3. I still take issue with the liberal use of the word "precise" throughout the manuscript. In their reply to me, the authors state that "by 'precise patterning' we mean that the HCs and SCs gets close to a checkerboard like pattern. We show that the number of SC neighbors gets close to 4 and that the HCs arrange in a pattern close to hexagonal pattern." My response is: how is it reasonable to say "close to" is

the same as "precise"? To avoid the semantic argument, you should say *how* precise if you want call something "precise." If the authors want to expend the effort quantifying their precision, then I encourage them to do so. Otherwise, from my read of the manuscript, they should not use the term so frequently or loosely.

We have replaced the term 'precise patterning' with the term 'ordered patterning' across the manuscript.

4. I appreciate the authors explaining their use of the term "compressibility." I am fine with them dropping "area" from "area compressibility." Even in light of this explanation, though, some of their description is still misleading. For example, the sentence that starts with, "The model predicted that since SCs are more compressible than HCs... ." From their explanation, SCs are NOT more compressible than HCs -- their apical areas are more compressible. The authors need to clean up this description to better reflect what they mean.

We have corrected the mentioned sentence to clarify that the *apical* side of SCs is more compressible than that of HCs.

5. The manuscript title is still an overstatement. The authors have not shown that what they are observing is shear-induced crystallization. They have shown that the patterning process is analogous to shear-induced crystallization. To show that the process is shear-induced crystallization, the authors would have to either apply a controlled shear stress by external means, or figure out how to measure the hypothesized cell-generated shear stress, then link it to the crystallization process. I understand this may be difficult or maybe impossible, but the indirect evidence shown by the authors is not sufficient -- and the other referees expressed the same concerns. And once again, I object to the use of the word "precise" without qualification, given what the authors have actually shown.

We think our data strongly indicates that shear induced crystallization drives the patterning in the inner ear. This conclusion is based on (1) The direct observation of shear motion in our movies (2) The gradual arrangement in crystal-like organization, and (3) a mathematical model that shows that global shear and local repulsion can reproduce the observed organization.

To address the reviewer's concerns we now suggest the following title:

"Shear-induced crystallization **during ordered** patterning of hair cells in the mammalian inner ear".

This change does not rule out other causes for the ordered patterning.

I hope the authors appreciate that my concerns can be addressed with small amounts of editing. I will be able to recommend publication once the manuscript is free of inaccurate terminology and overstatements of what has been found.

Appendix: Structure factor of a finite 2D hexagonal lattice

Let's define the two basis vectors for the lattice as:

$$\vec{b}_1 = a\hat{x}, \quad \vec{b}_2 = \frac{1}{2}a\hat{x} + \frac{\sqrt{3}}{2}a\hat{y}$$

where a is the lattice spacing. Given that the lattice size is $L_x = N_x a$ and $L_y = \frac{\sqrt{3}}{2} a N_y$ in the x and y directions respectively, the structure factor of the lattice can be calculated as follows:

$$(1) \quad S(\vec{q}) = \frac{1}{N} \sum_{n=1}^N \sum_{m=1}^N e^{-i\vec{q} \cdot (\vec{R}_n - \vec{R}_m)}$$

where \vec{R}_n/\vec{R}_m is the position vector of lattice point n/m , and $N \equiv N_x N_y$. Each of the sums in (1) can be broken into two axes such that $\sum_{n=1}^N [\dots] \equiv \sum_{n_x=1}^{N_x} \sum_{n_y=1}^{N_y} [\dots]$, and the position vectors can be written as $\vec{R}_n = n_x \vec{b}_1 + n_y \vec{b}_2$. Substituting into eq. (1) we get:

$$\begin{aligned} S(\vec{q}) &= \frac{1}{N_x N_y} \sum_{n_x=1}^{N_x} e^{-i\vec{q} \cdot \vec{b}_1 n_x} \sum_{n_y=1}^{N_y} e^{-i\vec{q} \cdot \vec{b}_2 n_y} \sum_{m_x=1}^{N_x} e^{i\vec{q} \cdot \vec{b}_1 m_x} \sum_{m_y=1}^{N_y} e^{i\vec{q} \cdot \vec{b}_2 m_y} = \\ &= \frac{1}{N_x N_y} \left| \sum_{n_x=1}^{N_x} e^{-i\vec{q} \cdot \vec{b}_1 n_x} \right|^2 \left| \sum_{n_y=1}^{N_y} e^{-i\vec{q} \cdot \vec{b}_2 n_y} \right|^2 = \frac{1}{N_x N_y} \left| \frac{1 - e^{-i\vec{q} \cdot \vec{b}_1 N_x}}{1 - e^{-i\vec{q} \cdot \vec{b}_1}} \right|^2 \left| \frac{1 - e^{-i\vec{q} \cdot \vec{b}_2 N_y}}{1 - e^{-i\vec{q} \cdot \vec{b}_2}} \right|^2 = \\ &= \frac{1}{N_x N_y} \left[\frac{\sin\left(\frac{\vec{q} \cdot \vec{b}_1 N_x}{2}\right)}{\sin\left(\frac{\vec{q} \cdot \vec{b}_1}{2}\right)} \right]^2 \left[\frac{\sin\left(\frac{\vec{q} \cdot \vec{b}_2 N_y}{2}\right)}{\sin\left(\frac{\vec{q} \cdot \vec{b}_2}{2}\right)} \right]^2 \end{aligned}$$

The domain size of the sample can be derived from the first zero of the structure factor along q_x (taking $q_y = 0$). Substituting $\vec{q} = (q_x, 0)$ and the basis vectors into the structure factor:

$$0 = \frac{1}{N_x N_y} \left[\frac{\sin\left(\frac{q_x a N_x}{2}\right)}{\sin\left(\frac{q_x a}{2}\right)} \right]^2 \left[\frac{\sin\left(\frac{q_x a N_y}{4}\right)}{\sin\left(\frac{q_x a N_y}{4}\right)} \right]^2$$

Given that $N_x \gg N_y$ (fits in our case), the first zero of the structure factor satisfies:

$$\sin\left(\frac{q_x a N_x}{2}\right) = 0 \rightarrow \frac{q_x a N_x}{2} = \pi \rightarrow L_x = N_x a = \frac{2\pi}{q_x}$$

The first zero of the structure factor presented in the manuscript (Fig. S1d) is measured at $q_x = 0.021 \mu\text{m}^{-1}$, which indicates on a domain size of:

$$L_x \cong 299 \mu\text{m}$$

Since the domain size fits the length of the analyzed section, we can conclude that this is a crystal with a domain size of at least $300 \mu\text{m}$.